# Demystifying amortized causal discovery with transformers

**Francesco Montagna**
**Institute of Science and Technology Austria & Chan Zuckerberg Initiative**
`francesco.montagna@ist.ac.at`

**Max Cairney-Leeming**
**Institute of Science and Technology Austria**

**Dhanya Sridhar**
**Mila - Quebec AI Institute and Université de Montréal**

**Francesco Locatello**
**Institute of Science and Technology Austria**

**Reviewed on OpenReview:** `https://openreview.net/forum?id=9Lgy7IGSfp`

## Abstract

Supervised learning for causal discovery from observational data often achieves competitive performance despite seemingly avoiding the explicit assumptions that traditional methods require for identifiability. In this work, we analyze CSIvA (Ke et al., 2023b) on bivariate causal models, a transformer architecture for amortized inference promising to train on synthetic data and transfer to real ones. First, we bridge the gap with identifiability theory, showing that the training distribution implicitly defines a prior on the causal model of the test observations: consistent with classical approaches, good performance is achieved when we have a good prior on the test data, and the underlying model is identifiable. Second, we find that CSIvA can not generalize to classes of causal models unseen during training: to overcome this limitation, we theoretically and empirically analyze *when* training CSIvA on datasets generated by multiple identifiable causal models with different structural assumptions improves its generalization at test time. Overall, we find that amortized causal discovery with transformers still adheres to identifiability theory, violating the previous hypothesis from Lopez-Paz et al. (2015) that supervised learning methods could overcome its restrictions.

## 1 Introduction

Causal discovery aims to uncover the underlying causal relationships between variables of a system from pure observations, which is crucial for answering interventional and counterfactual queries when experimentation is impractical or unfeasible (Peters et al., 2017; Pearl, 2009; Spirtes, 2010). Unfortunately, causal discovery is inherently ill-posed (Glymour et al., 2019): unique identification of causal directions requires restrictive assumptions on the class of structural causal models (SCMs) that generated the data (Shimizu et al., 2006; Hoyer et al., 2008; Zhang & Hyvärinen, 2009). These theoretical limitations often render existing methods inapplicable, as the underlying assumptions are usually untestable or difficult to verify in practice (Montagna et al., 2023a).

Recently, supervised learning algorithms trained on synthetic data have been proposed to overcome the need for specific hypotheses, which restrains the application of classical causal discovery methods to real-world problems (Ke et al., 2023b; Lopez-Paz et al., 2015; Li et al., 2020; Lippe et al., 2022; Lorch et al., 2022). Seminal work from Lopez-Paz et al. (2015) argues that this learning-based approach to causal discovery would allow dealing with

complex data-generating processes and would greatly reduce the need for explicitly crafting identifiability conditions a-priori: despite this ambitious goal, the output of these methods is generally considered unreliable, as no theoretical guarantee is provided. A pair of non-identifiable structural causal models can be associated with different causal directed acyclic graphs (DAGs) $\mathcal{G} \neq \tilde{\mathcal{G}}$, while entailing the same joint distribution $p$ on the system's variables. It is thus unclear how a learning algorithm presented with observational data generated from $p$ would be able to overcome these theoretical limits and correctly identify a unique causal structure. However, the available empirical evidence seems not to reflect impossibility results, as these methods yield surprising generalization abilities on several synthetic benchmarks. Our work aims to bridge this gap by studying the performance of a transformer architecture for causal discovery through the lens of the theory of identifiability from observational data. Specifically, we analyze the CSIvA (Causal Structure Induction via Attention) model for causal discovery (Ke et al., 2023b), focusing on bivariate graphs, as they offer a controlled yet non-trivial setting for the investigation. As our starting point, we provide closed-form examples that identify the limitations of CSIvA in recovering causal structures of linear non-Gaussian and nonlinear additive noise models, which are notably identifiable, and demonstrate the expected failures through empirical evidence. These findings suggest that the class of structural causal models that can be identified by CSIvA is inherently dependent on the specific class of SCMs observed during training. Thus, the need for restrictive hypotheses on the data-generating process is intrinsic to causal discovery, both in the traditional and modern learning-based approaches: assumptions on the test distribution either are posited when selecting the algorithm (traditional methods) or in the choice of the training data (learning-based methods). To address this limitation, we theoretically and empirically analyze *when* training CSIvA on datasets generated by multiple identifiable SCMs with different structural assumptions improves its generalization at test time. Our experimental findings are based on the analysis of ∼1M test runs. In summary:

- We empirically show that the class of structural causal models that CSIvA can identify is defined by the class of SCMs observed through samples during the training. We reinforce the notion that identifiability in causal discovery inherently requires assumptions, which must be encoded in the training data in the case of learning algorithms for amortized inference: this violates a previous hypothesis in Lopez-Paz et al. (2015), which suggests that these methods could exceed the boundaries of identifiability.

- We empirically show that CSIvA is expected to fail to generalize on datasets generated by structural causal models characterized by mechanism types or noise terms distributions unseen during training. While this appears as a significant limit of amortized causal discovery, systematic analysis has been disregarded by previous work in the literature.

- To mitigate this limitation, we study the benefits of CSIvA training on mixtures of causal models. We analyze when algorithms learned on multiple models are expected to identify broad classes of SCMs (unlike many classical methods). Empirically, we show that training on samples generated by multiple identifiable causal models with different assumptions on mechanisms and noise distribution results in significantly improved generalization abilities.

## 2 Related works

In this paper, we study *amortized inference of bivariate causal graphs*, i.e. supervised optimization of an inference model to directly predict a causal structure from newly provided data. In particular, this is the first work that draws a connection between identifiability theory and amortized inference of causal DAGs. Dai et al. (2023) studies supervised learning of the graph skeleton, limiting its analysis to the role of identifiability of unshielded triplets. Several algorithms have instead been proposed.

**Algorithms for amortized inference closely related to CSIvA.** In the context of purely observational data, Lopez-Paz et al. (2015) defines a classification problem mapping the kernel mean embedding of the data distribution to a causal graph, while Li et al. (2020) relies on equivariant neural network architectures. More recently, Lippe et al. (2022) and Lorch et al. (2022) proposed learning on interventional data, in addition to observations (in the same spirit as CSIvA). Despite different algorithmic implementations, the target object of estimation of most of these methods is the distribution over the space of all possible graphs, conditional on the

input dataset (similarly, the ENCO algorithm in Lippe et al. (2022) models the conditional distribution of individual edges). This justifies our choice of restricting our study to the CSIvA architecture (despite this being a clear limitation), as in the infinite observational sample limit, these methods approximate the same distribution.

**Other learning-based algorithms for causal discovery.** Out of the scope of this work, there are methods that necessarily require interventional data (Brouillard et al., 2020; Ke et al., 2023a; Scherrer et al., 2022), and learning-based algorithms unsuitable for amortized inference (Lachapelle et al., 2020; Ng et al., 2020; Zheng et al., 2018; Zhang et al., 2022; Bello et al., 2022).

**Differences with Lopez-Paz et al. (2015).** Before moving forward, we remark on the main differences between our paper and Lopez-Paz et al. (2015), as both works concentrate on supervised learning for the inference of *bivariate* causal graphs. Lopez-Paz et al. (2015) frames causal discovery as a classification problem, where the goal is estimating and mapping the kernel mean embeddings of the distribution of the observed data to the correct causal order (assuming a causal relation is in place). Building on the theory of reproducing kernel Hilbert spaces, they provide finite sample learning rates. In particular, their study assumes observations generated by identifiable causal models. In contrast, we aim to (i) empirically investigate what conditions enable identifiability in amortized causal discovery (ii) theoretically and empirically investigate how to exploit the known identifiability results to train algorithms with improved test generalization.

## 3 Background and motivation

We start introducing structural causal models (SCMs), an intuitive framework that formalizes causal relations. Let $X$ be a set of random variables in $\mathbb{R}$ defined according to the set of structural equations:

$$X_i := f_i(X_{\mathrm{PA}_i^{\mathcal{G}}}, N_i), \ \ \forall i = 1, \dots, k. \tag{1}$$

$N_i \in \mathbb{R}$ are *noise* random variables. The function $f_i$ is the *causal mechanism* mapping the set of *direct causes* $X_{\mathrm{PA}_i^{\mathcal{G}}}$ of $X_i$ and the noise term $N_i$, to $X_i$'s value. The *causal graph* $\mathcal{G}$ is a directed acyclic graph (DAG) with nodes $X = \{X_1, \dots, X_k\}$, and edges $\{X_j \to X_i : X_j \in X_{\mathrm{PA}_i^{\mathcal{G}}}\}$, with $\mathrm{PA}_i^{\mathcal{G}}$ indices of the parent nodes of $X_i$ in $\mathcal{G}$. The causal model induces a density $p_X$ over the vector $X$.

### 3.1 Causal discovery from observational data

Causal discovery from observational data is the inference of the causal graph $\mathcal{G}$ from a dataset of i.i.d. observations of the random vector $X$. In general, without restrictive assumptions on the mechanisms and the noise distributions, the direction of edges in the graph $\mathcal{G}$ is not identifiable, i.e. it can not be found from the population density $p_X$. In particular, it is possible to identify only a Markov equivalence class, which is the set of graphs encoding the same conditional independencies as the density $p_X$. To clarify with an example, consider the causal graph $X_1 \to X_2$ associated with a structural causal model inducing a density $p_{X_1, X_2}$. If the model is not identifiable, there exists an SCM with causal graph $X_2 \to X_1$ that entails the same joint density $p_{X_1, X_2}$. The set $\{X_1 \to X_2, X_2 \to X_1\}$ is the Markov equivalence class of the graph $X_1 \to X_2$, i.e. the set of all graphs with $X_1, X_2$ mutually dependent. Clearly, in this setting, even the exact knowledge of $p_{X_1, X_2}$ cannot inform us about the correct causal direction.

**Definition 1** (Identifiable causal model)**.** Consider a structural causal model with underlying graph $\mathcal{G}$ and $p_X$ joint density of the causal variables. We say that the model is *identifiable* from observational data if the density $p_X$ can not be entailed by a structural causal model with graph $\tilde{\mathcal{G}} \neq \mathcal{G}$.

We define the *post-additive noise model* (post-ANM) as the causal model with the set of equations:

$$X_i := f_{2,i}(f_{1,i}(X_{\mathrm{PA}_i^{\mathcal{G}}}) + N_i), \ \forall i = 1, \dots, d, \tag{2}$$

with $f_{2,i}$ invertible map and mutually independent noise terms. When $f_{2,i}$ is a nonlinear function, the post-ANM amounts to the identifiable *post-nonlinear* model (PNL) (Zhang & Hyvärinen, 2009). When $f_{2,i}$ is the identity function and $f_{1,i}$ nonlinear, it simplifies to the nonlinear *additive noise model* (ANM)(Hoyer

et al., 2008; Peters et al., 2014), which is known to be identifiable, and is described by the set of structural equations:

$$X_i := f_{1,i}(X_{\mathrm{PA}_i^{\mathcal{G}}}) + N_i. \tag{3}$$

If, additionally, we restrict the mechanisms $f_{1,i}$ to be linear and the noise terms $N_i$ to a non-Gaussian distribution, we recover the identifiable *linear non-Gaussian additive model* or LiNGAM (Shimizu et al., 2006):

$$X_i = \sum_{j \in \mathrm{PA}_i^{\mathcal{G}}} \alpha_j X_j + N_i, \quad \alpha_j \in \mathbb{R}. \tag{4}$$

The special case of linear Gaussian models, with noise terms with equal variance, is also known to be identifiable (Peters & Bühlmann, 2013).

## 3.2 Motivation and problem definition

Causal discovery from observational data relies on specific assumptions, which can be challenging to verify in practice (Montagna et al., 2023a). To address this, recent methods leverage supervised learning for the *amortized inference of causal graphs* (or simply *amortized causal discovery*), i.e. optimization of an inference model to directly predict a causal structure from newly provided data (Lopez-Paz et al., 2015; Li et al., 2020; Lippe et al., 2022; Lorch et al., 2022; Ke et al., 2023a; Löwe et al., 2020). While these approaches also aim to reduce reliance on explicit identifiability assumptions, they often lack a clear connection to the existing causal discovery theory, making their outputs generally unreliable. We illustrate this limitation through an example.

**Example 1.** We consider the CSIvA transformer architecture proposed by Ke et al. (2023b), which can learn a map from observational data to a causal graph. The authors of the paper show that, in the infinite sample regime, the CSIvA architecture exactly approximates the conditional distribution $p(\cdot|\mathcal{D})$ over the space of possible graphs, given a dataset $\mathcal{D}$. Identifiability theory in causal discovery tells us that if the class of structural causal models that generated the observations is sufficiently constrained, then there is only one graph that can fit the data within that class. For example, consider the case of a dataset that is known to be generated by a nonlinear additive noise model, and let $p(\cdot|\mathcal{D}, \mathrm{ANM})$ be the conditional distribution that incorporates this prior knowledge on the SCM: then $p(\cdot|\mathcal{D}, \mathrm{ANM})$ concentrates all the mass on a single point $\mathcal{G}^*$, the true graph underlying the $\mathcal{D}$ observations. Instead, in the absence of restrictions on the structural causal model, all the graphs in a Markov equivalence class are equally likely to be the correct solution given the data. Hence, $p(\cdot|\mathcal{D})$, the distribution learned by CSIvA, assigns equal probability to each graph in the Markov equivalence class of $\mathcal{G}^*$.

The claims made in our Example 1 are valid for all learning methods that approximate the conditional distribution over the space of graphs given the input data (Ke et al., 2023b; Lopez-Paz et al., 2015; Li et al., 2020; Lippe et al., 2022; Lorch et al., 2022), and suggest that these algorithms are at most informative about the equivalence class of the causal graph underlying the observations. However, the available empirical evidence does not seem to highlight these limitations, as in practice these methods can infer the true causal DAG on several synthetic benchmarks. Thus, further investigation is necessary if we want to rely on their output in any meaningful sense. In this work, we analyze these "black-box" approaches through the lens of established theory of causal discovery from observational data (causal inference often lacks experimental data, which we do not consider). We study in detail the CSIvA architecture (Ke et al., 2023b) (described in Appendix A), a variation of the transformer neural network (Vaswani et al., 2017) for the supervised learning of algorithms for amortized causal discovery. This model is optimized via maximum likelihood estimation, i.e. finding $\Theta$ that minimizes $-\mathbf{E}_{\mathcal{G},\mathcal{D}}[\ln \hat{p}(\mathcal{G}|\mathcal{D}; \Theta)]$, where $\hat{p}(\mathcal{G}|\mathcal{D}; \Theta)$ is the conditional distribution of a graph $\mathcal{G}$ given a dataset $\mathcal{D}$ parametrized by $\Theta$. We limit the analysis to CSIvA as it is a simple yet competitive end-to-end approach to learning causal models. While this is clearly a limitation of the paper, our theoretical and empirical conclusions exemplify both the role of theoretical identifiability in modern approaches and the new opportunities they provide. Additionally, it fits well within a line of works arguing that specifically transformers can learn causal concepts Jin et al. (2024); Zhang et al. (2024); Scetbon et al. (2024) and can be explicitly trained to identify different assumptions, such as linearity of the mechanisms or normality of the noise terms, when doing in-context learning (Gupta et al., 2023; Brown et al., 2020).

# 4 Experimental results through the lens of theory

In this section, we present a comprehensive analysis of bivariate causal discovery with transformers and its relation to the theoretical boundaries of causal discovery from observational data [1]. We show that suitable assumptions must be encoded in the training distribution to ensure the identifiability of the test data, and we additionally study the effectiveness of training on mixtures of causal models to overcome these limitations, improving generalization abilities. In Appendix C we discuss how our findings can be naturally extended to the case of multivariate causal models: the intuition is that, in this case, identifiability can be guaranteed by iteratively verifying that the causal order of all bivariate subgraphs is individually identifiable (Theorem 28 in Peters et al. (2014)). As a result, it is common to limit the analyses to bivariate graphs (see e.g. Hoyer et al. (2008); Zhang & Hyvärinen (2009); Immer et al. (2023); Xi et al. (2025)), which justifies our choice.

## 4.1 Experimental design

We concentrate our research on causal models of two variables, causally related according to one of the two graphs $X \rightarrow Y$, $Y \rightarrow X$. Bivariate models are the simplest non-trivial setting with a well-known theory of causality inference (Hoyer et al., 2008; Zhang & Hyvärinen, 2009; Peters et al., 2014), but also amenable to manipulation. This allows for comprehensive training and analysis of diverse SCMs and facilitates a clear interpretation of the results.

**Datasets.** Unless otherwise specified, in our experiments we train CSIvA on a sample of 15000 synthetically generated datasets, consisting of 1500 i.i.d. observations. Classes of SCMs are defined by the mechanism type and the noise terms distribution (e.g., linear non-Gaussian): each dataset is generated from a single SCM instance sampled from that class. The coefficients of the linear mechanisms are sampled in the range $[-3, -0.5] \cup [0.5, 3]$, removing small coefficients to avoid *close-to-unfaithful* effects (Uhler et al., 2012). Nonlinear mechanisms are parametrized according to a neural network with random weights, a strategy commonly adopted in the literature of causal discovery (see Appendix B.2; alternatively, we provide experiments on data generated simulating nonlinear mechanisms by sampling from a Gaussian process, as described in Appendix D.7). The post-nonlinearity of the PNL model consists of a simple map $z \mapsto z^3$. Noise terms are sampled from common distributions and a randomly generated density that we call *mlp*, previously adopted in Montagna et al. (2023a), defined by a standard Gaussian transformed by a multilayer perceptron (MLP) (Appendix B.2). We name these datasets *mechanism-noise* to refer to their underlying causal model. For example, data sampled from a nonlinear ANM with Gaussian noise are named *nonlinear-gaussian*. More details on the synthetic data generation schema are found in Appendix B.2. All data are standardized by their empirical variance to remove opportunities to learn shortcuts (Geirhos et al., 2020; Reisach et al., 2021; Montagna et al., 2023b).

**Metric and random baseline.** As our metric we use the structural Hamming distance (SHD), which is the number of edge removals, insertions or flips required to transform the predicted graph to the ground-truth. In the context of bivariate causal graphs with a single edge, this is simply an error count, so correct inference corresponds to SHD = 0, and an incorrect prediction gives SHD = 1. Additionally, we define a reference random baseline, which assigns a causal direction according to a fair coin, achieving SHD = 0.5 in expectation. Each architecture we analyze in the experiments is trained 3 times, with different parameter initialization and training samples: the SHD presented in the plots is the average of each of the 3 models on 1500 distinct test datasets of 1500 points each, and the error bars are 95% confidence intervals.

We detail the training hyperparameters in Appendix B.1. Next, we analyze our experimental results, starting by investigating how well CSIvA generalizes on distributions unseen during training.

## 4.2 Warm up: is CSIvA capable of in and out-of-distribution generalization?

**In-distribution generalization.** First, we investigate the generalization of CSIvA on datasets sampled from the structural casual model that generates the train distribution, with mechanisms and noise

---

[1]The code for CSIvA implementation can be found here. The code for reproducing the experiments of the paper can be found here

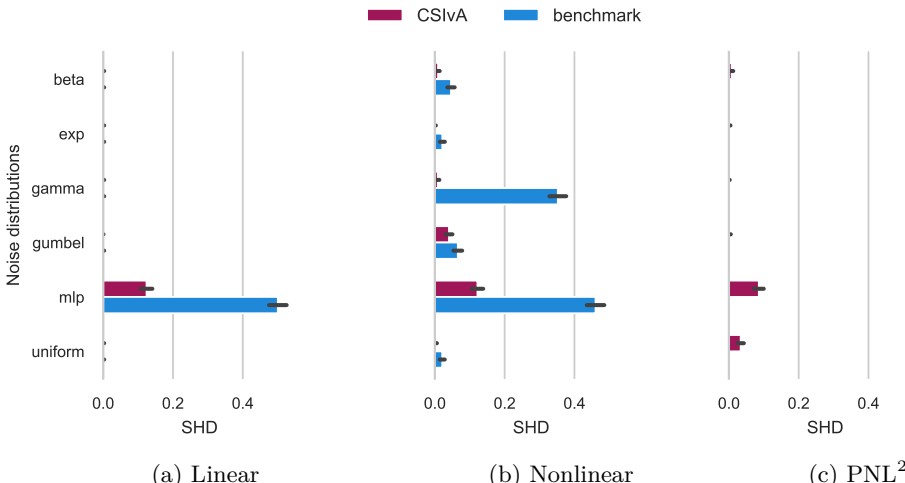

Figure 1: In-distribution generalization of CSIvA trained and tested on data generated according to the same structural causal models, fixing mechanisms, and noise distributions between training and testing. As baselines for comparison, we use DirectLiNGAM on linear SCMs and NoGAM on nonlinear ANM (we use their causal-learn and dodiscover implementations). CSIvA performance is clearly non-trivial and generalizing well.

distributions fixed between training and testing. We call this *in-distribution generalization*. The main goal of these experiments is to validate that the performance of our CSIvA implementation is non-trivial. As a benchmark, we present the accuracy of two state-of-the-art approaches from the literature on causal discovery: we consider the DirectLiNGAM and NoGAM algorithms (Shimizu et al., 2011; Montagna et al., 2023c), respectively designed for the inference on LiNGAM and nonlinear ANM generated data[2]. The results of Figure 1 show that CSIvA can properly generalize to unseen samples from the training distribution: the majority of the trained models present SHD close to zero and comparable to the relative benchmark algorithm.

**Out-of-distribution generalization.** In practice, we generally do not know the SCM defining the test distribution, so we are interested in CSIvA's ability to generalize to data sampled from a class of causal models that is unobserved during training. We call this *out-of-distribution generalization* (OOD). We study OOD generalization to different noise terms, analyzing the network performance on datasets generated from causal models where the mechanisms are fixed with respect to the training, while the noise distribution varies (e.g., given linear-mlp training samples, testing occurs on linear-uniform data). Orthogonally to these experiments, we empirically validate CSIvA's OOD generalization over different mechanism types (linear, nonlinear, post-nonlinear), while leaving the noise distribution (mlp) fixed across test and training. In Figure 2a, we observe that CSIvA cannot generalize across the different mechanisms, as the SHD of a network tested on unseen causal mechanisms approximates that of the random baseline. Further, Figure 2b shows that out-of-distribution generalization across noise terms does not work reliably, and it is hard to predict when it might occur. We note that these findings are novel in the literature: OOD experiments in the sense we define can not be found in Ke et al. (2023b); Li et al. (2020) empirical results are limited to the OOD generalization from linear Gaussian models to linear models with Exponential, Gumbel and Poisson noise. Lorch et al. (2022) analyses generalization from SCMs with Gaussian noise to Laplace and Cauchy distributions, and fixed mechanisms class. The remaining literature on amortized inference that we discuss in Section 2 generally disregards these experiments.

**Implications.** CSIvA generalizes well to test data generated by the same class of SCMs used for training, in line with the findings in Ke et al. (2023b), which validates our implementation and training procedure. However, it struggles when the test data are out-of-distribution, generated by causal models with different

---

[2]The causal-learn implementation of the PNL algorithm could not perform better than random on our synthetic post-nonlinear data, and we observed that this was due to the sensitivity of the algorithm to the variance scale. So we report the plot of Figure 1c without benchmark comparison. We remark that the point of this experiment is not to make any claims on CSIvA being state-of-the-art but to validate that the performance we obtain in our re-implementation is non-trivial. This is clear for PNL, even without comparison.

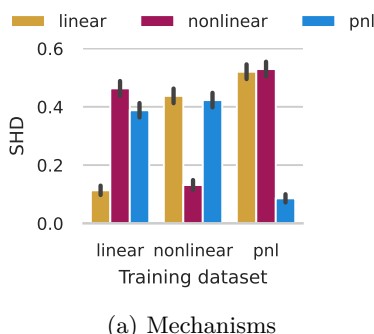
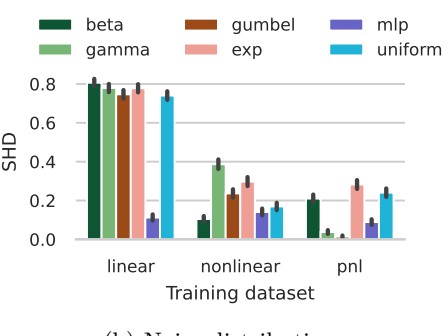

(a) Mechanisms          (b) Noise distributions

Figure 2: Out-of-distribution generalisation. We train three CSIvA models on data sampled from SCMs with linear, nonlinear additive, and post-nonlinear mechanisms; and fixed *mlp* noise distribution. In Figure 2a we test across different mechanism types, with mlp-distributed noise terms both in test and training. In Figure 2b we test across different noise distributions, with test mechanism types fixed from training. CSIvA struggles to generalize to unseen causal mechanisms and often displays degraded performance over new noise distributions.

mechanism types and noise distributions than those it was trained on. From a practical perspective, this is a relevant finding, given that existing works on amortized causal discovery lack systematic experiments on the OOD setting. While training on a wider class of SCMs might overcome this limitation, it requires caution. The identifiability of causal graphs indeed results from the interplay between the data-generating mechanisms and noise distribution. However, as we argue in our Example 1, the class of causal models that a supervised learning algorithm can identify is generally not clear. In what follows, we investigate this point and its implications for CSIvA, showing that the identifiability of the test samples can be ensured by imposing suitable assumptions on the class of SCMs generating the training distribution.

### 4.3 How does CSIvA relate to identifiability theory for causal graphs?

The CSIvA algorithm does not make structural assumptions about the causal model underlying the input data. This implies that the output of this method is unclear: as CSIvA should target the conditional distribution $p(\cdot|\mathcal{D})$ over the space of graphs, in the absence of restrictions on the functional mechanisms and the distribution of the noise terms, the causal graph $X \to Y$ is indistinguishable from $Y \to X$, as they are both equally likely to underlie the joint density $p_{X,Y}$ generating the data. As we discuss in Example 1, the graphical output of the trained architecture could at most identify the equivalence class of the true causal graph. Yet, our experiments of Section 4.2 show that CSIvA is capable of good in-distribution generalization, often inferring the correct DAG at test time. We explain this seeming contradiction with the following hypothesis, which motivates the experimental analysis in the remainder of this section.

> **Hypothesis (informal).** At test time, the datasets on which CSIvA is expected to generalize well are determined by the family of SCMs on which training occurred, and can be analyzed using the tools from identifiability theory.

Notably, if this hypothesis is verified, we can analyse when CSIvA is expected or not to work well; the remainder of this work empirically studies this claim. Before moving forward, we present the following example adapted from Hoyer et al. (2008), which supports and clarifies our statement.

**Example 2.** Consider the causal model $Y = f(X) + N$, where $f(X) = -X$ and $p_X, p_N$ are Gumbel densities $p_X(x) = \exp(-x - \exp(-x))$ and $p_N(n) = \exp(-n - \exp(-n))$. This model satisfies the assumptions of the LiNGAM, so it is identifiable, in the sense that a backward linear model (i.e., $Y \to X$ with linear mechanism $f$) with the same distribution does not exist. However, in this special case, we can build a backward nonlinear additive noise model $X = g(Y) + \tilde{N}$ with independent noise terms: taking $p_Y(y) = \exp(-y - 2\log(1 + \exp(-y)))$ to be the density of a logistic distribution, $p_{\tilde{N}}(\tilde{n}) = \exp(-2\tilde{n} - \exp(-\tilde{n}))$

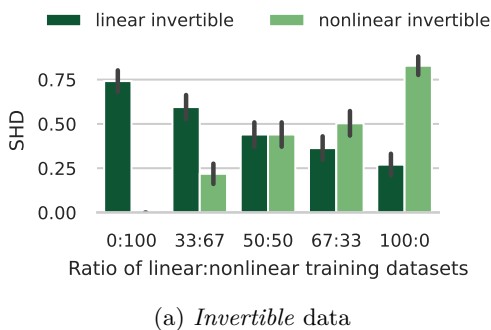
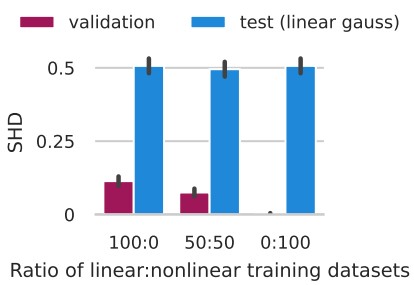

(a) *Invertible* data

(b) Linear Gaussian test data

Figure 3: Experiments on identifiability theory. Figure 3a shows the SHD of models trained on different ratios of *linear* and *nonlinear invertible* data of Example 2. In Figure 3b we test the performance on linear-Gaussian data. Models are trained with different ratios of samples from linear and nonlinear SCMs with Gaussian noise terms. The validation results showcase that the networks were trained successfully. In both cases, CSIvA behaves according to identifiability theory, failing to predict on *invertible* data (50:50 ratio) and linear Gaussian models.

and $g(y) = \log(1 + \exp(-y))$; we see that $p_{X,Y}$ can factorize according to two opposite causal directions, as $p_{X,Y}(x, y) = p_N(y - f(x))p_X(x) = p_{\bar{N}}(x - g(y))p_Y(y)$. Given a dataset $\mathcal{D}$ of observations from the forward linear model $X \to Y$, causal discovery methods like DirectLiNGAM (Shimizu et al., 2011) can provably identify the correct causal direction ($X \to Y$), assuming that sufficient samples are provided. Instead, the behavior of CSIvA seems hard to predict: given that the network approximates the conditional distribution $p(\cdot | \mathcal{D})$ over the possible graphs, for $\mathcal{D}$ with arbitrary many samples we have $p(X \to Y | \mathcal{D}) = p(Y \to X | \mathcal{D}) = 0.5$. On the other hand, given the prior knowledge that the data-generating SCM is a linear non-gaussian additive noise model, we have $p(X \to Y | \mathcal{D}, \text{LiNGAM}) = 1$, because the LiNGAM is identifiable. In this sense, the class of structural causal models that CSIvA correctly infers appears to be determined by the structural causal models underlying the generation of the training data. Under this hypothesis, training CSIvA exclusively on LiNGAM-generated data is equivalent to learning the distribution $p(\cdot | \mathcal{D}, \text{LiNGAM})$, such that the network should be able to identify the forward linear model, whereas it could only infer the equivalence class of the causal graph if its training datasets include observations from a nonlinear additive noise model.

The empirical results of Figure 3a show that CSIvA behaves according to our hypothesis: when training exclusively occurs on datasets $\{\mathcal{D}_{i,\to}\}_i$ generated by the *forward linear-gumbel model* of Example 2, the network can identify the causal direction of test data generated according to the same SCM. Similarly, the transformer trained on datasets $\{\mathcal{D}_{i,\leftarrow}\}_i$ from the *backward nonlinear model* of the example can generalize to test data coming from the same distribution. According to our claim, instead, the network that is trained on the union of the training samples $\{\mathcal{D}_{i,\to}\}_i \cup \{\mathcal{D}_{i,\leftarrow}\}_i$ from the forward and backward models (*50:50* ratio in Figure 3a) displays the same test SHD (around 0.5) as a random classifier assigning the causal direction with equal probability.

Further, we investigate CSIvA's relation with known identifiability theory by training and testing the architecture on data from a linear Gaussian model, which is well-known to be unidentifiable. Not surprisingly, the results of Figure 3b show that none of the algorithms that we learn can infer the causal order of linear Gaussian models with test SHD any better than a random baseline.

**Implications.** Our experiments show that CSIvA learns algorithms that closely follow identifiability theory for causal discovery. In particular, while the method itself does not require explicit assumptions on the data-generating process, the chosen training data ultimately determines the class of causal models identifiable during inference. Notably, previous work has argued that supervised learning approaches in causal discovery would help with "dealing with complex data-generating processes and greatly reduce the need of explicitly crafting identifiability conditions a-priori", Lopez-Paz et al. (2015). In the case of CSIvA, this expectation does not appear to be fulfilled, as the assumptions still need to be encoded explicitly in the training data. However, this observation opens two new important questions: (1) Can we train a single network to correctly classify multiple (or even all) identifiable causal structures? (2) How much ambiguity might exist between these identifiable models? We start by answering this second question.

### 4.4 An identifiability argument in favor of learning from multiple causal models

Example 2 of the previous section shows that elements of distinct classes of identifiable structural causal models, such as LiNGAM and nonlinear ANM, may become non-identifiable when we consider their union. In this section, we discuss the identifiability of the post-additive noise models. Previously, Hoyer et al. (2008) showed that the set of distributions generated according to the additive noise model equation 3 and that is non-identifiable is negligible. Later, Zhang & Hyvärinen (2009) characterized non-identifiable post-nonlinear models in terms of the properties of their functional mechanisms, and the distribution of the noise terms. In this section, we discuss how these results put together show that the set of distributions generated according to a post-ANM that is non-identifiable is negligible.

Let $X, Y$ be a pair of random variables generated according to the causal direction $X \rightarrow Y$ and the post-additive noise model structural equation:

$$Y = f_2(f_1(X) + N_Y), \tag{5}$$

where $N_Y$ and $X$ are independent random variables, and $f_2$ is invertible. If the SCM is non-identifiable, the data-generating process can be described by a *backward* model with the structural equation:

$$X = g_2(g_1(Y) + N_X), \tag{6}$$

$N_X$ independent from $Y$, and $g_2$ invertible. We introduce the random variables $\tilde{X}, \tilde{Y}$, such that the forward and backward equations can be rewritten as

$$Y = f_2(\tilde{Y}), \quad \tilde{Y} := f_1(X) + N_Y,$$
$$X = g_2(\tilde{X}), \quad \tilde{X} := g_1(Y) + N_X.$$

We note that equivalently the following invertible additive noise models on $\tilde{X}, \tilde{Y}$ hold:

$$\tilde{Y} = h_Y(\tilde{X}) + N_Y, \quad h_Y := f_1 \circ g_2, \tag{7}$$
$$\tilde{X} = h_X(\tilde{Y}) + N_X, \quad h_X := g_1 \circ f_2. \tag{8}$$

Equations (7) and (8) reduce the problem of studying the identifiability of a post-ANM to that of studying the identifiability of an additive noise model, as done in Theorem 1 of Hoyer et al. (2008), which we repropose in Appendix E: intuitively, the statement of the theorem says that the space of all continuous distributions generated according to a bivariate additive noise model and that is non-identifiable is contained in a 2-dimensional space. As the space of continuous distributions of random variables is infinite-dimensional, we conclude that the ANM is generally identifiable. Given that, according to Equation equation 7 and Equation equation 8, the post-ANM can be refactored in an additive noise model, the guarantees of identifiability still hold (for the formal statement and proof see Appendix E).

**Implications.** As we discussed, the post-ANM is generally identifiable, which suggests that the setting of Example 2 is rather artificial. This result provides the theoretical ground for training causal discovery algorithms on datasets generated from multiple identifiable SCMs. This is particularly appealing in the case of CSIvA, given the poor OOD generalization ability observed in our experiments of Section 4.2.

### 4.5 Can we train CSIvA on multiple causal models for better generalization?

In this section, we investigate the benefits of training over multiple causal models, i.e. on samples generated by a combination of classes of identifiable SCMs characterized by different mechanisms and noise terms distribution. Our motivation is as follows: given that our empirical evidence shows that CSIvA is capable of in-distribution generalization, whereas dramatically degrades the performance when testing occurs out-of-distribution, it is thus desirable to increase the class of causal models represented in the training datasets. We separately study the effects of training over multiple mechanisms and multiple noise distributions and compare the testing performance against architectures trained on samples of a single SCM.

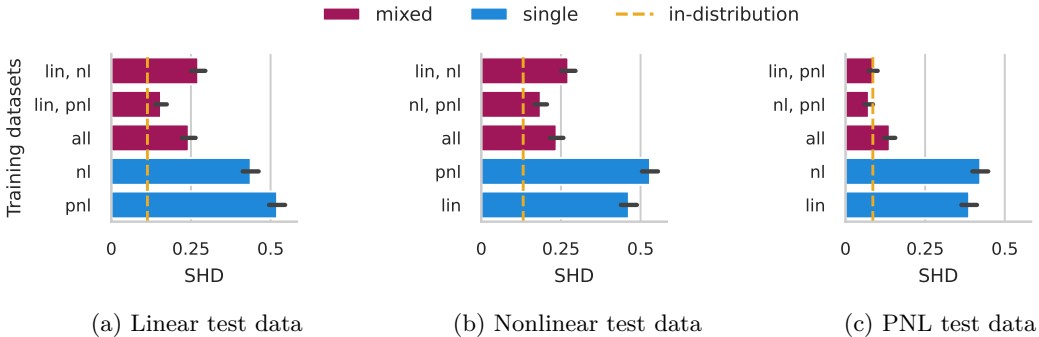

Figure 4: Mixture of causal mechanisms. We train four models on samples from structural casual models with different mechanism types. We compare their test SHD (the lower, the better) against networks trained on datasets generated according to a single type of mechanism. The dashed line indicates the test SHD of a model trained on samples with the same mechanisms as test SCM. Training on multiple causal models with different mechanisms (*mixed* bars) always improves performance compared to training on single SCMs.

**Mixture of causal mechanisms.** We consider four networks optimized by training of CSIvA on datasets generated from pairs (or triples) of distinct SCMs, with fixed *mlp* noise and which differ in terms of their mechanisms type: linear and nonlinear; nonlinear and post-nonlinear; linear and post-nonlinear; linear, nonlinear and post-nonlinear. The number of training datasets for each architecture is fixed (15000) and equally split between the causal models with different mechanism types. The results of Figure 4 show that the networks trained on mixtures of mechanisms all present significantly better test SHD compared to CSIvA models trained on a single mechanism type. We find that learning on multiple SCMs improves the SHD from $\sim 0.5$ to $\sim 0.2$ both on linear and nonlinear test data (Figures 4a and 4b), and even better accuracy is achieved on post-nonlinear samples, as shown in Figure 4c.

**Mixture of noise distributions.** Next, we analyze the test performance of three CSIvA networks optimized on samples from structural causal models that have different distributions for their noise terms, while keeping the mechanism types fixed. Figure 5 shows that training over different noises (beta, gamma, gumbel, exponential, mlp, uniform) always results in a network that is agnostic with respect to the noise distributions of the SCM generating the test samples, always achieving SHD $< 0.1$, with the exception of datasets with mlp error terms (0.2 average SHD on linear and nonlinear data).

**Implications.** We have shown that learning on mixtures of SCMs with different noise term distributions and mechanism types leads to models generalizing to a much broader class of structural causal models during testing. Hence, combining datasets generated from multiple models looks like a promising framework to overcome the limited out-of-distribution generalization abilities of CSIvA observed in Section 4.2. However, it is easier to incorporate prior assumptions on the class of causal mechanisms (linear, non-linear, post-non-linear) compared to the noise distributions (which are potentially infinite). This introduces a trade-off between amortized inference and classical methods for causal discovery: for example, RESIT, NoGAM, AdaScore, and LNMIX (Peters et al., 2014; Montagna et al., 2023c; 2025; Liu et al., 2024) algorithms require no assumptions on the noise type, but only work for a limited class of mechanisms (linear and nonlinear with additive noise).

## 5 Conclusion

In this work, we investigate the interplay between identifiability theory and supervised learning for amortized inference of causal graphs, using CSIvA as the ground of our study. Consistent with classical algorithms, we demonstrate that good performance can be achieved if (i) we have a good prior on the structural causal model generating the test data (ii) the setting is identifiable. In particular, prior knowledge of the test distribution is encoded in the training data in the form of constraints on the structural causal model underlying their generation. With these results, we highlight the need for identifiability theory in modern learning-based approaches to causality, while past works have mostly disregarded this connection. Further, our findings

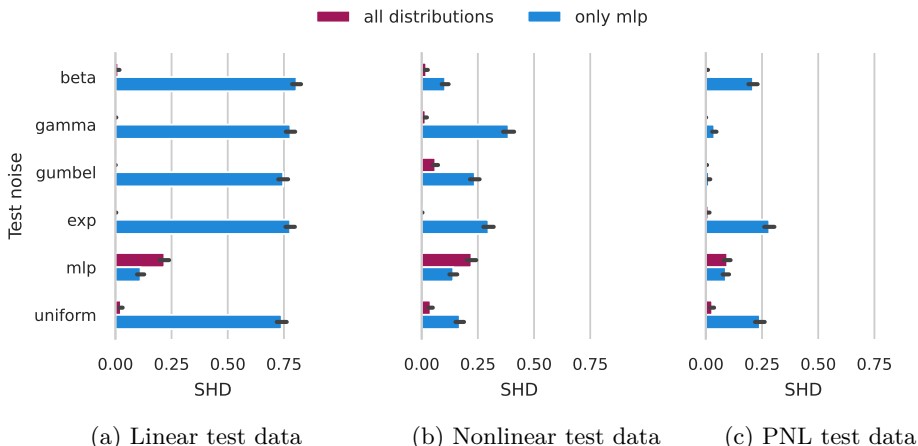

Figure 5: Mixture of noise distributions. We train three networks on samples from SCMs with different noise terms distributions and fixed mechanism types: linear, nonlinear, and post-nonlinear. We present their test SHD (the lower, the better) on data from SCMs with the mechanisms fixed with respect to training, and noise terms changing between each dataset. Training on multiple causal models with different noises (*all distributions* bars) always improves performance compared to training on single SCMs with fixed mlp noise (*only mlp* bars).

provide the theoretical ground for training on observations sampled from multiple classes of identifiable SCMs, a strategy that improves test generalization to a broad class of causal models. Finally, we highlight an interesting new trade-off regarding identifiability: traditional methods like LiNGAM, RESIT, and PNL require strong restrictions on the structural mechanisms underlying the data generation (linear, nonlinear additive, or post-nonlinear) while generally being agnostic relative to the noise terms distribution. Training on mixtures of causal models instead offers an alternative that is less reliant on assumptions on the mechanisms, while incorporating knowledge about all possible noise distributions in the training data is practically impossible to achieve. We leave it to future work to reproduce our analysis on a wider class of architectures, as well as extend our study to interventional data with more than two nodes.

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

# A  Learning to induce: causal discovery with transformers

## A.1  A supervised learning approach to causal discovery

First, we describe the training procedure for the CSIvA architecture, which aims to learn the distribution of causal graphs conditioned on observational and/or interventional datasets. We omit interventional datasets from the discussion as they are not of interest to our work. Training data are generated from the joint distribution $p_{\mathcal{G},\mathcal{D}}$ between a graph $\mathcal{G}$ and a dataset $\mathcal{D}$. First, we sample a set of directed acyclic graphs $\{\mathcal{G}^i\}_{i=1}^n$ with nodes $X_1, \ldots, X_d$, from a distribution $p_{\mathcal{G}}$. Then, for each graph we sample a dataset of $m$ observations of the graph nodes $\mathcal{D}^i = \{x_1^j, \ldots, x_d^j\}_{j=1}^m$, $i = 1, \ldots, n$. Hence, we build a training dataset $\{\mathcal{G}^i, \mathcal{D}^i\}_{i=1}^n$.

The CSIvA model defines a distribution $\hat{p}_{\mathcal{G}|\mathcal{D}}(\cdot; \Theta)$ of graphs conditioned on the observational data and parametrized by $\Theta$. Given an invertible map $\mathcal{G} \mapsto A$ from a graph to its binary adjacency matrix representation of $d \times d$ entries (where $A_{ij} = 1$ iff $X_i \to X_j$ in $\mathcal{G}$), we consider an equivalent estimated distribution $\hat{p}_{A|\mathcal{D}}(\cdot; \Theta)$, which has the following autoregressive form:

$$\hat{p}_{A,\mathcal{D}}(A|\mathcal{D}; \Theta) = \prod_{l=1}^{d^2} \sigma(A_l; \rho = f_\Theta(A_1, \ldots, A_{l-1}, \mathcal{D})),$$

where $\sigma(\cdot; \rho)$ is a Bernoulli distribution parametrized by $\rho$. $\rho$ itself is a function of $f_\Theta$ defined by the encoder-decoder transformer architecture, taking as input previous elements of the matrix $A$ (here represented as a vector of $d^2$ entries) and the dataset $\mathcal{D}$. $\Theta$ is optimized via maximum likelihood estimation, i.e. $\Theta^* = \arg\min_\Theta -\mathbf{E}_{\mathcal{G},\mathcal{D}}[\ln \hat{p}(\mathcal{G}|\mathcal{D}; \Theta)]$, which corresponds to the usual cross-entropy loss for the Bernoulli distribution. Training is achieved using stochastic gradient descent, in which each gradient update is performed using a pair $(\mathcal{D}^i, A^i)$, $i = 1 \ldots, d$. In the infinite sample limit, we have $\hat{p}_{\mathcal{G}|\mathcal{D}}(\cdot; \Theta^*) = p_{\mathcal{G}|\mathcal{D}}(\cdot)$, while in the finite-capacity case, it is only an approximation of the target distribution.

## A.2  CSIvA architecture

In this section, we summarize the architecture of CSIvA, a transformer neural network that can learn a map from data to causally interpreted graphs, under supervised training.

**Transformer neural network.**  Transformers (Vaswani et al., 2017) are a popular neural network architecture for modeling structured, sequential data data. They consist of an *encoder*, a stack of layers that learns a representation of each element in the input sequence based on its relation with all the other sequence's elements, through the mechanism of self-attention, and a decoder, which maps the learned representation to the target of interest. Note that data for causal discovery are not sequential in their nature, which motivates the adaptations introduced by Ke et al. (2023b) in their CSIvA architecture.

**CSIvA embeddings.**  Each element $x_i^j$ of an input dataset is embedded into a vector of dimensionality $E$. Half of this vector is allocated to embed the value $x_i^j$ itself, while the other half is allocated to embed the unique identity for the node $X_i$. We use a node-specific embedding because the values of each node may have very different interpretations and meanings. The node identity embedding is obtained using a standard 1D transformer positional embedding over node indices. The value embedding is obtained by passing $x_i^j$, through a multi-layer perceptron (MLP).

**CSIvA alternating attention.**  Similarly to the transformer's encoder, CSIvA stacks a number of identical layers, performing self-attention followed by a nonlinear mapping, most commonly an MLP layer. The main difference relative to the standard encoder is in the implementation of the self-attention layer: as transformers are in their nature suitable for the representation of sequences, given an input sample of $D$ elements, self-attention is usually run across all elements of the sequence. However, data for causal discovery are tabular, rather than sequential: one option would be to unravel the $n \times d$ matrix of the data, where $n$ is the number of observations and $d$ the number of variables, into a vector of $n \cdot d$ elements, and let this be the input sequence of the encoder. CSIvA adopts a different strategy: the self-attention in each encoder layer consists of alternate passes over the attribute and the sample dimensions, known as *alternating attention* Kossen et al.

| Hypeparameter | Value |
|---|---|
| Hidden state dimension | 64 |
| Encoder transformer layers | 8 |
| Decoder transformer layers | 8 |
| Num. attention heads | 8 |
| Optimizer | Adam |
| Learning rate | $10^{-4}$ |
| Samples per dataset ($n$) | 1500 |
| Num. training datasets | 15000 |
| Num. iterations | $< 150000$ |
| Batch size | 5 |

Table 1: Hyperparameters for the training of the CSIvA models of the experiments in Section 4.

(2021). As a clarifying example, consider a dataset $\{(x_1^i, x_2^i)\}_{i=1}^n$ of $n$ i.i.d. samples from the joint distribution of the pair of random variables $X_1, X_2$. For each layer of the encoder, in the first step (known as *attention between attributes*), attention operates across all nodes of a single sample $(x_1^i, x_2^i)$ to encode the relationships between the two nodes. In the second step (*attention between samples*), attention operates across all samples $(x_k^1, \ldots, x_k^n), k \in \{1, 2\}$ of a given node, to encode information about the distribution of single node values.

**CSIvA encoder summary.** The encoder produces a summary vector $s_i$ with $H$ elements for each node $X_i$, which captures essential information about the node's behavior and its interactions with other nodes. The summary representation is formed independently for each node and involves combining information across the $n$ samples. This is achieved with a method often used with transformers that involves a weighted average based on how informative each sample is. The weighting is obtained using the embeddings of a summary "sample" $n + 1$ to form queries, and embeddings of node's samples $\{x_i^j\}_{j=1}^n$ to provide keys and values, and then using standard key-value attention.

**CSIvA decoder.** The decoder uses the summary information from the encoder to generate a prediction of the adjacency matrix $A$ of the underlying $\mathcal{G}$. It operates sequentially, at each step producing a binary output indicating the prediction $\hat{A}_{i,j}$ of $A_{i,j}$, proceeding row by row. The decoder is an autoregressive transformer, meaning that each prediction $\hat{A}_{i,j}$ is obtained based on all elements of $A$ previously predicted, as well as the summary produced by the encoder. The method does not enforce acyclicity, although Ke et al. (2023b) shows that in cyclic outputs genereally don't occur, in practice.

## B  Training details

### B.1  Hyperparameters

In Table 1 we detail the hyperparameters of the training of the network of the experiments. We define an iteration as a gradient update over a batch of 5 datasets. Models are trained until convergence, using a patience of 5 (training until five consecutive epochs without improvement) on the validation loss - this always occurs before the 25-th epoch (corresponding to $\approx 150000$ iterations). The batch size is limited to 5 due to memory constraints.

### B.2  Synthetic data

In this section, we provide additional details on the synthetic data generation, which was performed with the `causally`[3] Python library (Montagna et al., 2023a). Our data-generating framework follows that of Montagna et al. (2023a), an extensive benchmark of causal discovery methods on different classes of SCMs.

---

[3]`https://causally.readthedocs.io/en/latest/`

**Distribution of the noise terms.** We generated datasets from structural causal models with the following distribution of the noise terms: Beta, Gamma, Gaussian (for nonlinear data), Gumbel, Exponential, and Uniform. Additionally, we define the *mlp* distribution by nonlinear transformations of gaussian samples from a guassian distribution centered at zero and with standard deviation $\sigma$ uniformly sampled between 0.5 and 1. The nonlinear transformation is parametrized by a neural network with one hidden layer with 100 units, and sigmoid activation function. The weights of the network are uniformly sampled in the range $[-1.5, 1.5]$. We additionally standardized the output of each *mlp* sample by the empirical variance computed over all samples.

**Causal mechanisms.** The *nonlinear mechanisms* of the PNL model and the nonlinear ANM model are generated by a neural network with one hidden layer with 10 hidden units, with a parametric ReLU activation function. The network weights are randomly sampled according to a standard Gaussian distribution (we refer to data with nonlinear mechanisms sampled according to this approach as *NN-data*). The *linear mechanisms* are generated by sampling the regression coefficients in the range $[-3, -0.5] \cup [0.5, 3]$.

**NN-data generation: literature review.** We present an extensive list of works adopting neural networks for the sampling of nonlinear mechanisms, similarly to our work: Brouillard et al. (2020; 2021); Lippe et al. (2022); Bello et al. (2022); Montagna et al. (2023a;b); Ke et al. (2023a;b); Reizinger et al. (2023); Massidda et al. (2023); Tran et al. (2024). This suggests that our data generation strategy is established in the literature of causality. Additional experiments with sampling of nonlinear mechanisms from Gaussian processes are presented in Appendix D.7.

Data are standardized with their empirical variance, which removes the presence of shortcuts which could be learned by the network, notably *varsortability* (Reisach et al., 2021) and *score-sortability* (Montagna et al., 2023b).

### B.3 Computer resources

Our experiments were run on a local computing cluster, using any and all available GPUs (all NVIDIA). For replication purposes, GTX 1080 Ti's are entirely suitable, as the batch size was set to match their memory capacity, when working with bivariate graphs. All jobs ran with 10GB of RAM and 4 CPU cores. The results presented in this paper were produced after 145 days of GPU time, of which 68 were on GTX 1080 Ti's, 13 on RTX 2080 Ti's, 11 on A10s, 19 on A40s, and 35 on RTX 3090s. Together with previous experiments, while developing our code and experimental design, we used 376 days of GPU time (for reference, at a total cost of 492.14 Euros), similarly split across whichever GPUs were available at the time: 219 on GTX 1080 Ti's, 38 on RTX 2080 Ti's, 18 on A10s, 63 on RTX 3090s, 31 on A40s, and 6 on A100s.

## C  CSIvA identifiability properties on multivariate SCMs

In the main manuscript, we limit our empirical and theoretical analysis of the identifiability guarantees provided by CSIvA to the case of bivariate causal models. In this section, we show how our findings are expected to extend to the multivariate setting. Our starting point is Theorem 28 from Peters et al. (2014): the intuition is that identifiability of multivariate additive noise models can be guaranteed by iteratively verifying that the causal order of all bivariate subgraphs is individually identifiable. We formalize this reporting the following set of definitions and results from Peters et al. (2014).

**Condition 1** (Condition 19 of Peters et al. (2014)). *Consider an additive noise model with structural equations $X_2 = f(X_1) + N$, $X_1, N$ independent random variables. The triple $(f, p_{X_1}, p_N)$ does not solve the following differential equation for all pairs $x_1, x_2$ with $f'(x_2)\nu''(x_2 - f(x_1)) \neq 0$:*

$$\xi''' = \xi'' \left( \frac{f''}{f'} - \frac{\nu''' f'}{\nu''} \right) + \frac{\nu''' \nu' f'' f'}{\nu''} - \frac{\nu'(f'')^2}{f'} - 2\nu'' f'' f' + \nu' f''', \tag{9}$$

*Here, $\xi := \log p_{X_1}$, $\nu := \log p_N$, the logarithms of the strictly positive densities. The arguments $x_2 - f(x_1)$, $x_1$, and $x_1$ of $\nu$, $\xi$ and $f$ respectively, have been removed to improve readability.*

The intuition is that a bivariate additive noise model (which can be seen as a reparametrization of a post-nonlinear model, as shown in our Section 4.4) is identifiable if it has a density that satisfies the above Condition 1. This can be generalized to the case of multivariate ANMs where, for identifiability to hold, Condition 1 must be verified for each pair of causally related variables in the SCM: when this is verified, we refer to a *restricted* additive noise model.

**Definition 2** (Definition 27 of Peters et al. (2014)). Consider an additive noise model with structural equations $X_i := f_i(X_{\mathrm{PA}_i^{\mathcal{G}}}) + N_i, i = 1, \ldots, k$, independent noise terms and causal graph $\mathcal{G}$. We call this SCM a *restricted additive noise model* if for all $X_j \in X$, $X_i \in X_{\mathrm{PA}_j^{\mathcal{G}}}$, and all sets $X_S \subseteq X$, $S \subset \mathbb{N}$, with $X_{\mathrm{PA}_j^{\mathcal{G}}} \setminus \{X_i\} \subseteq X_S \subseteq X_{\mathrm{ND}_j}^{\mathcal{G}} \setminus \{X_i, X_j\}$ (where $\mathrm{ND}_j$ is the set of non descendants of the node $X_j$ in the graph $\mathcal{G}$), there is a value $x_S$ with $p(x_S) > 0$, such that the triplet

$$(f_j(x_{\mathrm{PA}_j^{\mathcal{G}} \setminus \{i\}}, \cdot), p_{X_i | X_S = x_S}, p_{N_j})$$

satisfies Condition 1. Here, $f_j(x_{\mathrm{PA}_j^{\mathcal{G}} \setminus \{i\}}, \cdot)$ denotes the mechanism function $x_i \mapsto f_j(x_{\mathrm{PA}_j^{\mathcal{G}}})$. Additionally, we require the noise variables to have positive densities and the functions $f_j$ to be continuous and three times continuously differentiable.

In the above definition we adopted the following notation: for a random vector $X = (X_1, \ldots, X_n)$, and a set $S \subseteq \{1, \ldots, n\}$, we define $X_S$ as the vector with elements $\{X_i : i \in S\}$.

Finally, the next theorem formalizes the intuition we've advocated so far: the *restricted* additive noise model of Definition 2, i.e. an SCM whose pairwise causal relations are individually identifiable, is itself identifiable.

**Theorem 1** (Theorem 28 of Peters et al. (2014)). *Let $X$ be generated by a restricted additive noise model with graph $\mathcal{G}$, and assume that the causal mechanisms $f_j$ are not constant in any of the input arguments, i.e. for $X_i \in X_{\mathrm{PA}_j^{\mathcal{G}}}$, there exist $x_i \neq x_i'$ such that $f_j(x_{\mathrm{PA}_j^{\mathcal{G}} \setminus \{i\}}, x_i) \neq f_j(x_{\mathrm{PA}_j^{\mathcal{G}} \setminus \{i\}}, x_i')$. Then, $\mathcal{G}$ is identifiable.*

We note that this relation between bivariate and multivariate identifiability was recently exploited for causal discovery with optimal transport by Tu et al. (2022)

**Discussion and multivariate identifiability guarantees of transformers.** The above theorem states that a restricted ANM is identifiable. According to our Definition 2, an additive noise model $X_h := f_h(X_{\mathrm{PA}_h^{\mathcal{G}}}) + N_h, h = 1, \ldots, k$ is *restricted* if each pair of connected nodes $X_i \to X_j$, for each $X_{\mathrm{PA}_j^{\mathcal{G}}} \setminus \{X_i\} \subseteq X_S \subseteq X_{\mathrm{ND}_j}^{\mathcal{G}} \setminus \{X_i, X_j\}$ (think of $X_S$ as the set of all possible causes of $X_j$, except $X_i$), can define a bivariate SCM of the form $(f_j(x_{\mathrm{PA}_j^{\mathcal{G}} \setminus \{i\}}, \cdot), p_{X_i | X_S = x_S}, p_{N_j})$ that satisfies Condition 1, i.e. that is identifiable. How does this relate to our findings? Our experiments and analysis of Section 4.3 validate the hypothesis that transformers align with the theory of identifiability in the case of training and inference on bivariate graphs. Given Theorem 1, we know that multivariate identifiability is a property of SCMs where each pair of causes and effects can define a bivariate structural causal model that is itself identifiable: this implies that the empirical guarantees of identifiability we verify for transformers (via CSIvA) on bivariate models must extend to multivariate models. This is apparent by contradiction: say we train a CSIvA architecture that can infer the causal direction of a multivariate linear Gaussian model (which is notoriously non-identifiable). This means that our algorithm can infer the causal direction for each bivariate subgraph consisting of two variables connected according to a linear Gaussian structural equation: this would contradict our experimental results presented in Fig. 3b and analyzed in Section 4.3.

# D   Further experiments

In this section, we provide additional experiments on real-world data, on the scaling properties of CSIvA in the number of training samples, and benchmark CSIvA performance in comparison to several well-established or state-of-the-art methods for causal discovery, with identifiability guarantees under different assumptions: DirectLiNGAM (Shimizu et al., 2011) for inference on linear non-Gaussian models, CAM (Bühlmann et al., 2014) for inference of additive noise models with additive mechanisms, NoGAM (Montagna et al., 2023c) and GraNDAG (Lachapelle et al., 2020) for inference on ANMs (the latter is taken from the *continuous* optimization literature of causal discovery, already mentioned in the related works Section 2).

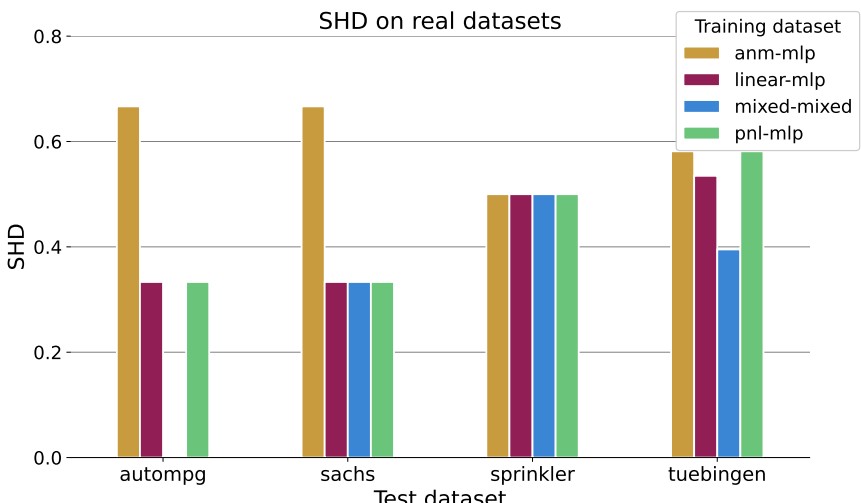

Figure 6: Average SHD (the lower, the better) on real-world datasets of CSIvA models that are trained on synthetic datasets generated with linear, nonlinear additive, and post-nonlinear mechanisms and fixed mlp noise distribution (linear-mlp, anm-mlp, pnl-mlp bars) and *mixed* mechanisms and *mixed* noise distributions (mixed-mixed bar). Performance is tested on bivariate models. We observe that the model optimized with mixed training is on par or outperforms the other algorithms.

### D.1  Experiments on real-world datasets

We consider the accuracy of CSIvA trained on different dataset configurations and tested on real-world datasets. In particular, we perform evaluation on the Tübingen pairs dataset (Mooij et al., 2016), the Sachs biological dataset (Sachs et al., 2005), the AutoMPG dataset on cars fuel consumption (Bache & Lichman, 2013) and the Sprinkler dataset, a simple dataset on the causal relations between the binary categorical variables `rain`, `sprinkler on/off`, `wet grass`. Given that our algorithms are trained on bivariate models, from each multivariate dataset we extract all possible two variables subgraphs where this operation does not introduce new confounding effects. This results in 9 datasets from Sachs, 3 datasets from AutoMPG, and 2 datasets from Sprinkler. We consider 102 pairs from the Tübingen dataset.

**Real-world generalization of mixed-trained models.**  In Fig. 6 we illustrate the average accuracy per dataset type (Sachs, AutoMPG, Sprinkler, Tübingen) of each CSIvA model. In particular, we want to probe the goodness of mixed training in real-world scenarios. To this end, we train four architectures on the following dataset configurations: linear-mlp, anm-mlp, pnl-mlp, *mixed-mixed*, where the latter denotes the model trained on SCMs with linear, additive nonlinear and post-nonlinear mechanisms, and Beta, Gamma, Gumbel, Exponential, MLP, and Uniform noise distributions. We find the following interesting outcome: the mixed-mixed architecture is on par with the others on the Sprinkler and the Sachs datasets and outperforms the other methods on the AutoMPG and the Tübingen pairs datasets. Despite these results must be taken cautiously, they provide evidence of a strong result, that mixed training appears to be beneficial even in real-world scenarios, those of actual interest in applications.

**Benchmark with classic causal discovery.**  We probe CSIvA test generalization in comparison with DirectLiNGAM, CAM, NoGAM, and GraNDAG methods. According to Fig. 7, interestingly we find that the mixed-mixed CSIvA model (trained on SCMs with linear, additive nonlinear and post-nonlinear mechanisms, and Beta, Gamma, Gumbel, Exponential, MLP, Uniform noise distributions) matches with or outperforms the other methods on all the test tasks. This provides additional empirical evidence on the benefits of the mixed training procedure we propose to achieve better test generalization.

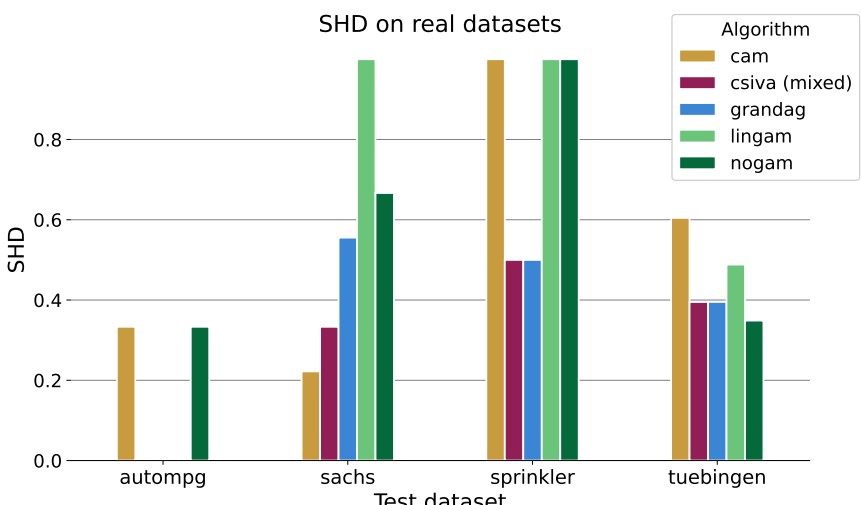

Figure 7: Average SHD (the lower, the better) on real-world datasets. The CSIvA model is trained on synthetic datasets generated with *mixed* mechanisms and *mixed* noise distributions (*csiva (mixed)* bar). As benchmark methods, we consider DirectLiNGAM, CAM, NoGAM, and GraNDAG. Performance is tested on bivariate models. We observe that the model optimized with mixed training is on par or outperforms the other algorithms.

## D.2    Benchmarking CSIvA generalization with classical causal discovery algorithms

In this section, we analyze the results of Fig. 8, where we compare the CSIvA trained on mixed mechanisms (linear, nonlinear, post-nonlinear) and mixed noises (all noise except for Gaussian) with the benchmark methods DirectLiNGAM, CAM, NoGAM, GraNDAG. Given that we want to probe CSIvA test generalization, we run inference over the following dataset configurations: linear-mixed (i.e. SCMs considering all possible noise distributions, except for Gaussian), anm-mixed, pnl-mixed, and mixed-mlp (i.e. SCMs with linear, nonlinear, post-nonlinear mechanisms). Fig. 8 shows results in line with our expectations: DirectLiNGAM, CAM, NoGAM, and GraNDAG achieve their best accuracy on data generated by SCMs respecting their assumptions, while degrading their performance on the other models; the CSIvA architecture trained on a mixture of SCMs with different mechanisms and noise distributions matches with or tops all other methods, in all the considered settings (while being outperformed on the anm and linear data, CSIvA still retains good average SHD accuracy).

## D.3    Experiments with different sizes of the training dataset

In this section, we explore how CSIvA test generalization scales when training occurs on different numbers of training samples. In the experiments on the main manuscript, each algorithm is optimized on 15000 datasets, where each dataset and the underlying causal graph corresponds to a training data point. We now compare the test SHD when training occurs on 5000 and 10000 datasets. One clear point emerges from the results of Fig. 9, that is our results on the benefits of the mixed training procedure are consistent for each size of the training dataset we considered. Moreover, we note that the performance of CSIvA does not appear to degrade due to the decrease in the number of training points.

## D.4    Can we learn to infer causal order from linear Gaussian data?

We ask whether CSIvA trained on non-identifiable models can implicitly learn to predict the causal direction of identifiable SCMs. For this purpose, we consider CSIvA optimized on linear Gaussian data and test its performance on several datasets sampled from structural causal models with different configurations of mechanisms and noise distributions: linear-mixed (with noise terms sampled according to all distributions except for Gaussian), anm-mixed, pnl-mixed, mixed-mlp (with mechanisms generated according to linear, nonlinear, post-nonlinear equations), anm-gauss, and pnl-gauss. The results of Fig. 10 present strong evidence

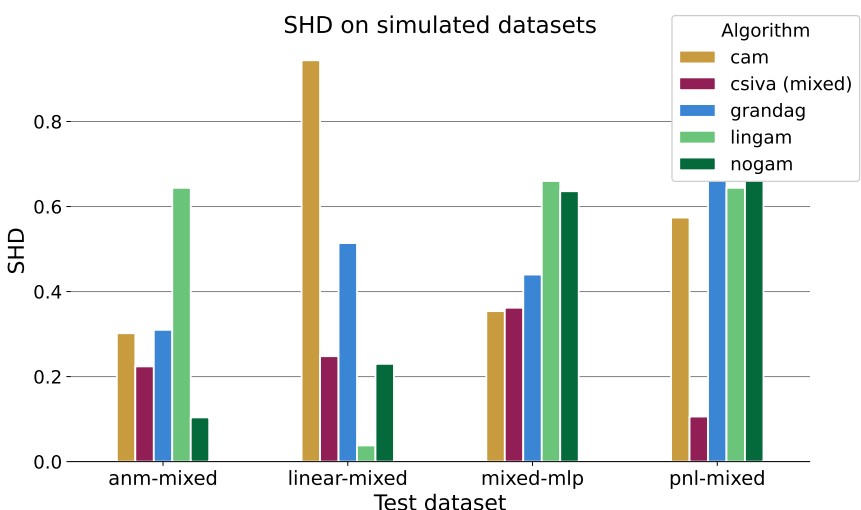

Figure 8: Average SHD (the lower, the better) on simulated datasets. The CSIvA model is trained on synthetic datasets generated with *mixed* mechanisms and *mixed* noise distributions (*csiva (mixed)* bar). As benchmark methods, we consider DirectLiNGAM, CAM, NoGAM, and GraNDAG. Performance is tested on bivariate models. We observe that, in general, the model optimized with mixed training is on par or outperforms the other algorithms.

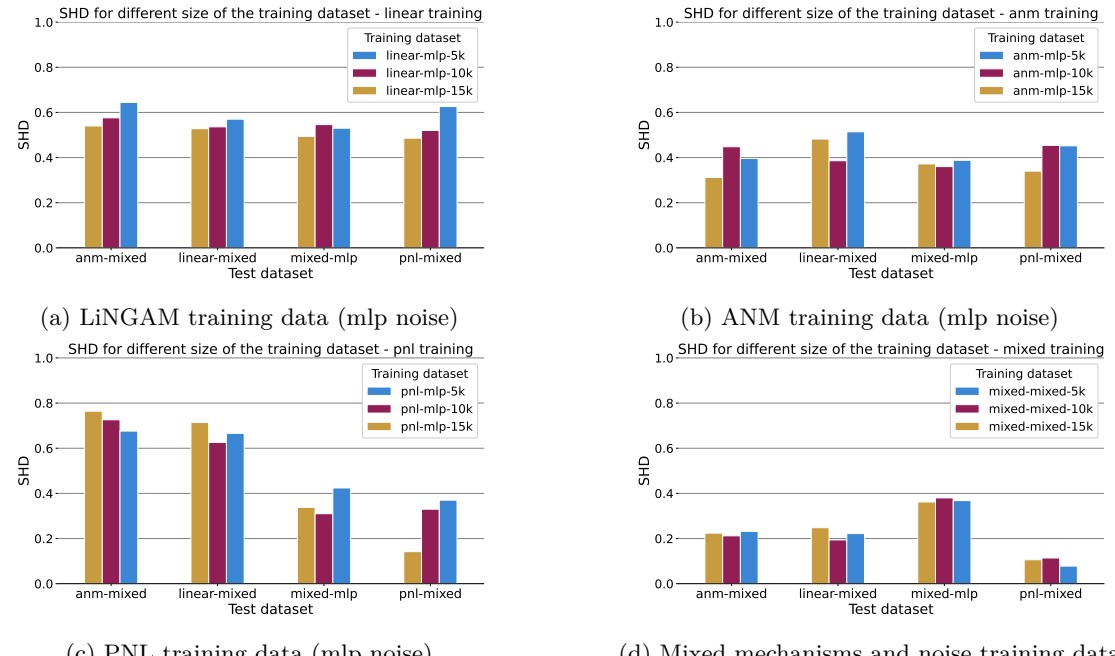

Figure 9: Average SHD (the lower, the better) of CSIvA models trained with 5000, 10000, 15000 data points. The algorithms are tested on simulated datasets generated with linear, nonlinear and post-nonlinear mechanisms (linear-mixed, anm-mixed, pnl-mixed entries on the x axis) and mixed-mlp datasets, generated with *mixed* mechanism types and fixed mlp noise distribution. We observe that (i) the mixed training improves the test generalization, irrespective of the training dataset size; (ii) CSIvA maintains its performance stable across different training dataset sizes.

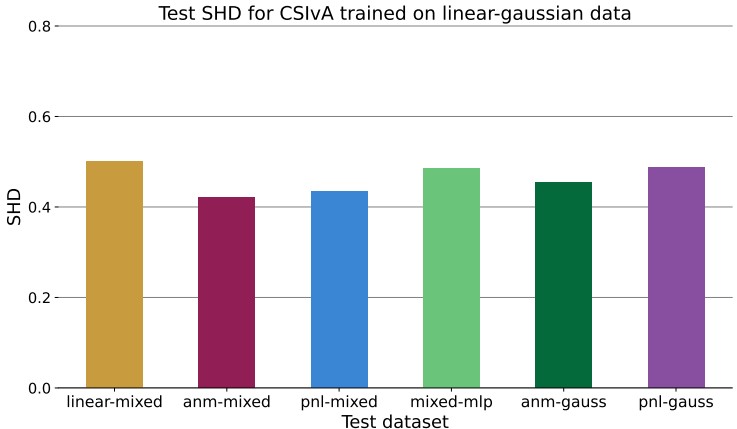

Figure 10: Average SHD (the lower, the better) of CSIvA trained on datasets generated by linear Gaussian models, which are non-identifiable. Performance is tested on simulated datasets generated according to several SCM configurations. We observe that training on non-identifiable data yields an algorithm that performs with average accuracy of 0.5, equivalent to a coin flip random baseline, across all the test tasks.

that models trained on non-identifiable SCMs can not infer the causal order: in fact, we see that consistently across all datasets CSIvA average SHD approximates 0.5, the performance of classification with a coin flip. This is in line with our expectations. In agreement with our motivating hypothesis (Section 4), in Section 4.3 we have empirically shown that CSIvA can model the class of the SCM generating the observed data and exploit this information to infer the correct causal DAG (instead of a less specific Markov equivalence class) when this is identifiable. Moreover, in our Section 4.2 our experiments show that CSIvA can not generalize to SCM classes unseen during training. In light of these findings, it is intuitive that an architecture trained on non-identifiable linear Gaussian data can only try to fit a linear Gaussian model, irrespective of the input data. Then, when inferring the causal direction, given that CSIvA assumes the data to be generated according to a linear Gaussian SCM, both the forward and backward directions are equally plausible, which explains the observed SHD close to 0.5.

### D.5 Experiments with bivariate independent graphs

In the main manuscript, we consider training and testing of CSIvA on bivariate graphs with an edge: $X \to Y$, $Y \to X$. This can be phrased as a classification problem with two labels. We motivate our choice by noticing that, in the bivariate setting, identifiability is a property of connected graphs: the empty graph with no edge defines a Markov equivalence class with one element, i.e. a singleton. This is known to be identifiable without explicit assumptions on the functional form of the mechanisms or the noise term distributions in the causal model. The goal of this section is to show that, if we include datasets generated according to an empty graph in the training procedure, CSIvA can learn to disambiguate between the three classes (the empty graph, $X \to Y$, $Y \to X$). To motivate our claim, we notice that classifying empty and connected graphs can be done by testing independence between the input variables: previous works phrase independence testing as a classification task and show that this can be learned via deep neural networks (Bellot & van der Schaar, 2019; Sen et al., 2017). The experiments of Fig. 11 sustain our claim. We consider three CSIvA architectures, each trained on independent pairs and one between linear, nonlinear, or post-nonlinear data. Our results show that the neural network can learn to disambiguate between the three classes in all scenarios.

### D.6 Mixed training with unlimited budget

We present our experimental results on one further question, to help clarify the results in the main text of the paper. We aim to understand when to make tradeoffs between computational resources, and having models that have been trained on a wider variety of SCMs. We compare training on multiple SCMs to single-SCM training, when all models see the same amount of training data from each SCM type as a non-mixed model

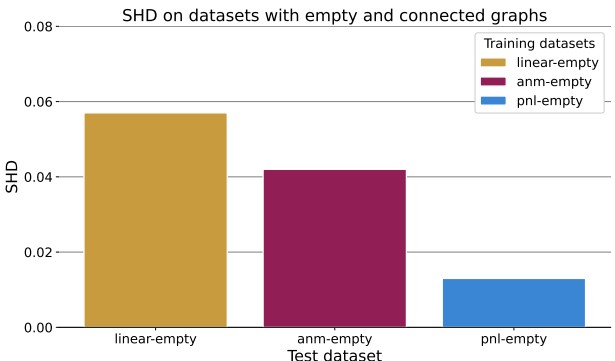

Figure 11: Average SHD (the lower, the better) for CSIvA trained on independent pairs and one between linear, nonlinear, or post-nonlinear data. Each algorithm is tested on the same class of structural causal models it was trained on. We note that in all three scenarios, CSIvA learns to distinguish all classes with almost optimal accuracy (i.e., SHD close to 0.

(i.e. a mixed network trains on $15,000$ linear datasets and $15,000$ PNL datasets, instead of $15,000$ divided between the two SCM types).

In the main text of this paper, we compare neural networks trained on a mix of structural causal models (e.g. noise distributions, or mechanism types), to models trained on a single mechanism-noise combination, where all models have the same amount of training data, $15,000$ datasets. In mixed training, we split these evenly, so a "lin, nl" model is trained on $7,500$ datasets from linear SCMs, and $7,500$ from nonlinear SCMs. Our results in this framework are promising, and show that for many combinations of SCM types, we can train one model instead of two, and achieve good progress, while making a $50\%$ savings on training costs. However, if our training budget is high/unlimited, we should also ask whether we can we achieve the same performance as a model trained on a single SCM type. Fig. 12 shows good results in this direction - the models trained with the same number of datasets per SCM type as an unmixed model had similar (or even better, for PNL data) performance as the un-mixed model trained on the same SCM type as the test data. These mixed models are also significantly more useful than having 2 or 3 separate models per SCM type, as they have good across-the-board performance. However, if we used the same computational resources to train 3 separate networks (one for each mechanism type) and wanted to use them for causal discovery on a dataset with unknown assumptions, we would be left with the rather difficult task of deciding which model to trust.

### D.7 Experiments with Gaussian process nonlinear mechanisms

In this section we present results obtained training and testing CSIvA on synthetic data with nonlinear mechanisms sampled from a Gaussian process with a unit bandwidth RBF kernel (we call data generated according to this approach as *GP-data*). In particular, for each variable $X_i$ node of the graph $\mathcal{G}$ generated according to model equation 1 we define the nonlinear mechanism $f_i(X_{\mathrm{PA}_i^{\mathcal{G}}}) = \mathcal{N}(\mathbf{0}, K(X_{\mathrm{PA}_i^{\mathcal{G}}}, X_{\mathrm{PA}_i^{\mathcal{G}}}))$, a multivariate normal distribution centered at zero and with covariance matrix as the Gaussian kernel $K(X_{\mathrm{PA}_i^{\mathcal{G}}}, X_{\mathrm{PA}_i^{\mathcal{G}}})$, where $X_{\mathrm{PA}_i^{\mathcal{G}}}$ are the observations of the parents of the node $X_i$. Together with our strategy adopted in the experiments in the main text of parametrizing nonlinearities with random neural networks (*NN-data*), this is one of the most common approaches in the literature.

**GP-data generation: literature review.** We present an extensive list of works adopting Gaussian processes for the sampling of nonlinear mechanisms: Rolland et al. (2022); Montagna et al. (2023a;b;c); Bühlmann et al. (2014); Mooij et al. (2016); Lachapelle et al. (2020); Wang et al. (2021); Chen et al. (2023); Mooij et al. (2011); Monti et al. (2019). This suggests that our data generation strategy is established in the causality literature.

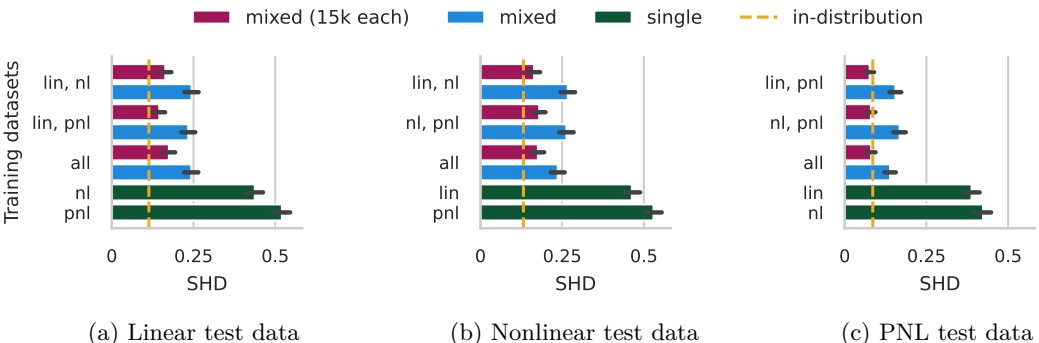

Figure 12: Mixtures of causal mechanisms, with varying amounts of training data. We train eight models on samples from structural casual models with different mechanisms. Four (in purple), were trained on $15,000$ samples for each SCM type (so the "lin,nl" model saw $30,000$ samples in total, and the "all" model saw $45,000$), and the other four (blue) are the same as in Fig. 4, and were trained on $15,000$ samples in total, evenly split between the SCM types they were trained on. We compare their test SHD (the lower, the better) against networks trained on datasets generated according to a single type of mechanism. The dashed line indicates the test SHD of a model trained on samples with the same mechanisms as the test SCM. Training on multiple causal models with different mechanisms (mixed bars) always improves performance compared to training on single SCMs.

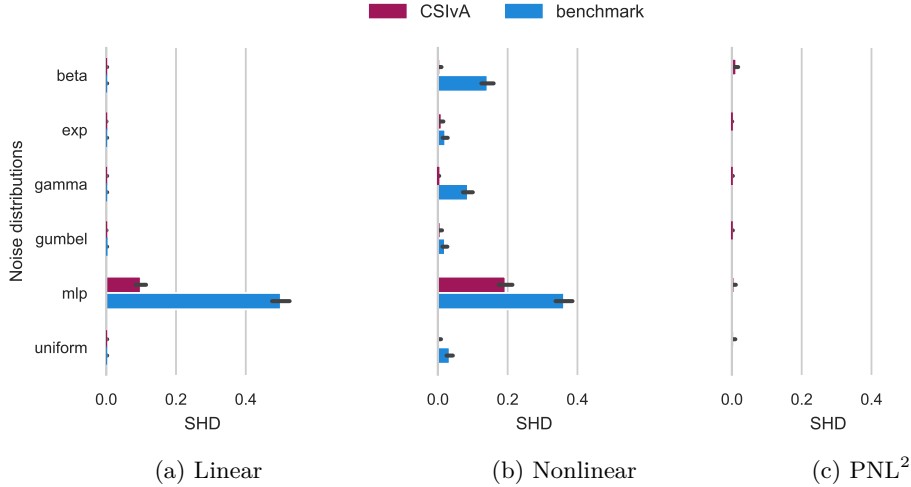

Figure 13: In-distribution generalization (GP-data) of CSIvA trained and tested on data generated according to the same structural causal models, fixing mechanisms, and noise distributions between training and testing. Nonlinear mechanisms for *nonlinear* and *pnl* data are sampled from a Gaussian process. As baselines for comparison, we use DirectLiNGAM on linear SCMs and NoGAM on nonlinear ANM (we use their causal-learn and dodiscover implementations). CSIvA performance is clearly non-trivial and generalizing well.

**Summary of the GP-data experiments.** Figures from 13 to 16 replicate the main text experiments involving nonlinear mechanisms either in the training or testing data. The results on GP-data agree with our findings on NN-data: CSIvA still shows poor OOD generalization under different training and test mechanisms, and generally for different training and testing noise distribution (except for PNL data). Similar to the case with NN-data, test generalization improves under mixed training.

# E   Theoretical results and proofs

In this section, we state and prove the identifiability of the post-ANM discussed in Section 4.3, as a corollary of Theorem 1 of Hoyer et al. (2008). The forward and backward models of equations equation 5 and equation 6

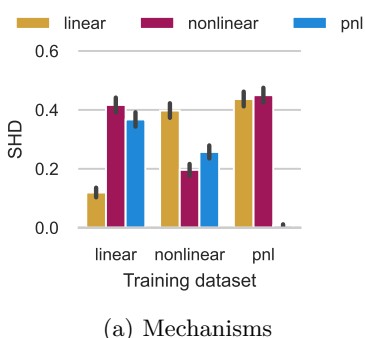

(a) Mechanisms

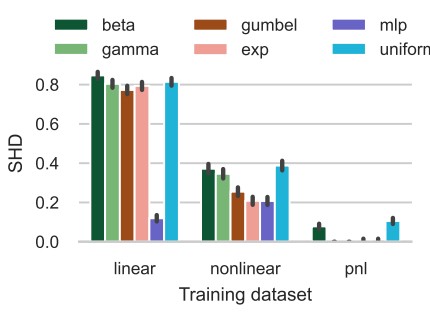

(b) Noise distributions

Figure 14: Out-of-distribution generalisation (GP-data). We train three CSIvA models on data sampled from SCMs with linear, nonlinear additive, and post-nonlinear mechanisms; and fixed *mlp* noise distribution. Nonlinear mechanisms for *nonlinear* and *pnl* data are sampled from a Gaussian process. In Figure 14a we test across different mechanism types, with mlp-distributed noise terms both in test and training. In Figure 14b we test across different noise distributions, with test mechanism types fixed from training. CSIvA struggles to generalize to unseen causal mechanisms and often displays degraded performance over new noise distributions.

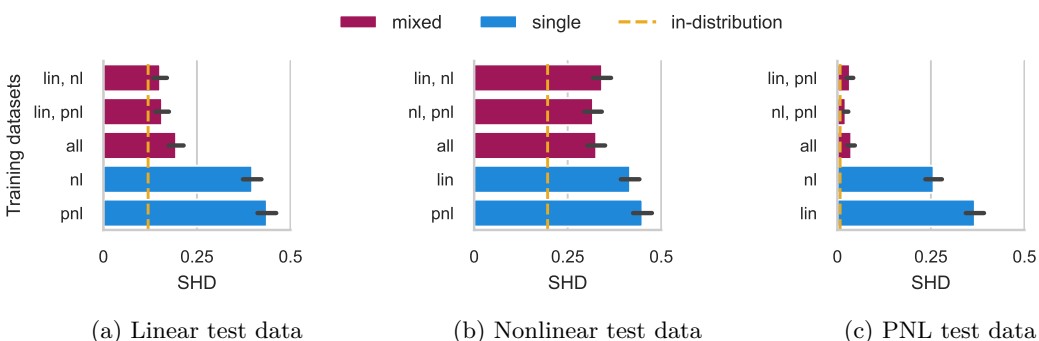

(a) Linear test data

(b) Nonlinear test data

(c) PNL test data

Figure 15: Mixture of causal mechanisms (GP-data). We train four models on samples from structural casual models with different mechanism types. Nonlinear mechanisms for *nonlinear* and *pnl* data are sampled from a Gaussian process. We compare their test SHD (the lower, the better) against networks trained on datasets generated according to a single type of mechanism. The dashed line indicates the test SHD of a model trained on samples with the same mechanisms as test SCM. Training on multiple causal models with different mechanisms (*mixed* bars) always improves performance compared to training on single SCMs.

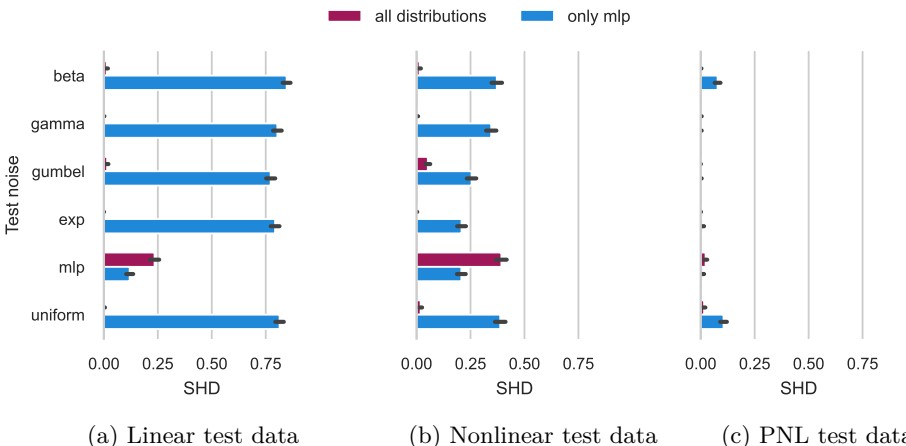

Figure 16: Mixture of noise distributions (GP-data). We train three networks on samples from SCMs with different noise terms distributions and fixed mechanism types: linear, nonlinear, and post-nonlinear. Nonlinear mechanisms for *nonlinear* and *pnl* data are sampled from a Gaussian process. We present their test SHD (the lower, the better) on data from SCMs with the mechanisms fixed with respect to training, and noise terms changing between each dataset. Training on multiple causal models with different noises (*all distributions* bars) always improves performance compared to training on single SCMs with fixed mlp noise (*only mlp* bars).

for the pair of random variables $X, Y$ is given by:

$$Y = f_2(f_1(X) + N_Y) = f_2(\tilde{Y}), \quad \tilde{Y} := f_1(X) + N_Y,$$
$$X = g_2(g_1(Y) + N_X) = g_2(\tilde{X}), \quad \tilde{X} := g_1(Y) + N_X,$$

with $f_2, g_2$ invertible functions, $N_Y, X$ independent random variables, and $N_X, Y$ independent random variables. Equivalently, we can frame forward and backward causal models for $\tilde{X}, \tilde{Y}$, as in equations equation 7 and equation 8:

$$\tilde{Y} = h_Y(\tilde{X}) + N_Y, \quad h_Y := f_1 \circ g_2,$$
$$\tilde{X} = h_X(\tilde{Y}) + N_X, \quad h_X := g_1 \circ f_2.$$

We are now ready to provide our identifiability statement for post-ANMs.

**Proposition 1** (Corollary of Theorem 1 of Hoyer et al. (2008)). *Let $p_{N_Y}, h_X, h_Y$ be fixed, and define $\nu_Y := \log p_{N_Y}, \xi := \log p_{\tilde{X}}$. Suppose that $p_{N_Y}$ and $p_{\tilde{X}}$ are strictly positive densities, and that $\nu_Y, \xi, f_1, f_2, g_1$, and $g_2$ are three times differentiable. Further, assume that for a fixed pair $h_Y, \nu_Y$ exists $\tilde{y} \in \mathbb{R}$ s.t. $\nu_Y''(\tilde{y} - h_Y(\tilde{x}))h_Y'(\tilde{x}) \neq 0$ is satisfied for all but a countable set of points $\tilde{x} \in \mathbb{R}$. Then, the set of all densities $p_{\tilde{X}}$ of $\tilde{X}$ such that both equations equation 5 and equation 6 are satisfied is contained in a 2-dimensional space.*

Before stating the proof of Proposition 1, we show under which condition the pair of random variables $X, Y$ satisfies the forward and backward models of equations equation 5, equation 6: this is relevant for our discussion, as the proof of Proposition 1 consists of showing that this condition is *almost* never satisfied.

**Notation.** We adopt the following notation: $\nu_X := \log p_{N_X}$, $\nu_Y := \log p_{N_Y}$, $\xi := \log p_{\tilde{X}}$, $\eta := \log p_{\tilde{Y}}$, and $\pi := \log p_{\tilde{X}, \tilde{Y}}$.

**Theorem 2** (Theorem 1 of Zhang & Hyvärinen (2009)). *Assume that $X, Y$ satisfies both causal relations of equations equation 5 and equation 6. Further, suppose that $p_{N_Y}$ and $p_{\tilde{X}}$ are positive densities on the support of $N_Y$ and $\tilde{X}$ respectively, and that $\nu_Y, \xi, f_1, f_2, g_1$, and $g_2$ are third order differentiable. Then, for each pair $(\tilde{x}, \tilde{y})$ satisfying $\nu_Y''(\tilde{y} - h_Y(\tilde{x}))h_Y(\tilde{x}) \neq 0$, the following differential equation holds:*

$$\xi''' = \xi'' \left( \frac{h_Y''}{h_Y'} - \frac{\nu_Y'''h_Y'}{\nu_Y''} \right) + \frac{\nu_Y'''\nu_Y'h_Y''h_Y'}{\nu_Y''} - \frac{\nu_Y'(h_Y'')^2}{h_Y'} - 2\nu_Y''h_Y''h_Y' + \nu_Y'h_Y''',$$

and $h_X$ is constrained in the following way:

$$\frac{1}{h'_X} = \frac{\xi'' + \nu''_Y(h'_Y)^2 - \nu'_Y h''_Y}{\nu''_Y h'_Y}, \tag{10}$$

*where the arguments of the functions have been left out for clarity.*

*Proof of Theorem 2.* We demonstrate separately the two statements of the theorem.

**Part 1.** Given that equations equation 5 and equation 6 hold, this implies that the forward and backward models on $\tilde{X}, \tilde{Y}$ of equations equation 7 and equation 8 are also valid, namely that:

$$\tilde{Y} = h_Y(\tilde{X}) + N_Y,$$
$$\tilde{X} = h_X(\tilde{Y}) + N_X.$$

These are the structural equations of two causal models, associated with the *forward* $\tilde{X} \to \tilde{Y}$ and *backward* $\tilde{Y} \to \tilde{X}$ graphs, respectively. Applying the Markov factorization of the distribution according to the forward direction, we get:

$$p_{\tilde{X},\tilde{Y}}(\tilde{x}, \tilde{y}) = p_{\tilde{Y}|\tilde{X}}(\tilde{y}|\tilde{x})p_{\tilde{X}}(\tilde{x}) = p_{N_Y}(\tilde{y} - h_Y(\tilde{x}))p_{\tilde{X}}(\tilde{x}),$$

which implies

$$\pi(\tilde{x}, \tilde{y}) = \nu_Y(\tilde{y} - h_Y(\tilde{x})) + \xi(\tilde{x}), \tag{11}$$

for any $\tilde{x}, \tilde{y}$. Similarly, the Markov factorization on the backward model implies:

$$\pi(\tilde{x}, \tilde{y}) = \nu_X(\tilde{x} - h_X(\tilde{y})) + \eta(\tilde{y}). \tag{12}$$

From equation 12, we have that:

$$\frac{\partial^2}{\partial \tilde{x}^2} \pi(\tilde{x}, \tilde{y}) = \nu''_X(\tilde{x} - h_X(\tilde{y}))$$
$$\frac{\partial^2}{\partial \tilde{x} \partial \tilde{y}} \pi(\tilde{x}, \tilde{y}) = -\nu''_X(\tilde{x} - h_X(\tilde{y}))h'_X(\tilde{y}),$$

which implies

$$\frac{\partial}{\partial \tilde{x}}\left( \frac{\frac{\partial^2}{\partial \tilde{x}^2}\pi(\tilde{x}, \tilde{y})}{\frac{\partial^2}{\partial \tilde{x} \partial \tilde{y}}\pi(\tilde{x}, \tilde{y})} \right) = 0. \tag{13}$$

Computing the same set of partial derivatives from equation 11, we find:

$$\frac{\partial^2}{\partial \tilde{x}^2} \pi(\tilde{x}, \tilde{y}) = \nu''_Y(\tilde{y} - h_Y(\tilde{x}))(h'_Y(\tilde{x}))^2 - \nu'_Y(\tilde{y} - h_Y(\tilde{x}))h''_Y(\tilde{x}) + \xi''(\tilde{x})$$
$$\frac{\partial^2}{\partial \tilde{x} \partial \tilde{y}} \pi(\tilde{x}, \tilde{y}) = -\nu''_Y(\tilde{y} - h_Y(\tilde{x}))h'_Y(\tilde{x}).$$

from which follows:

$$\frac{\partial}{\partial \tilde{x}}\left( \frac{\frac{\partial^2}{\partial \tilde{x}^2}\pi(\tilde{x}, \tilde{y})}{\frac{\partial^2}{\partial \tilde{x} \partial \tilde{y}}\pi(\tilde{x}, \tilde{y})} \right) = -2h''_Y + \frac{\nu'_Y h'''_Y}{\nu''_Y h'_Y} - \frac{\xi'''}{\nu''_Y h'_Y} + \frac{\nu'''_Y \nu'_Y h''_Y}{(\nu''_Y)^2} - \frac{\nu'_Y(h''_Y)^2}{\nu''_Y(h'_Y)^2} + \frac{\xi'' \nu'''_Y h''_Y}{(\nu''_Y)^2 \nu''_Y(h'_Y)^2}$$
$$= 0.$$

where we drop the input arguments for conciseness. The equality with 0 is given by the equality with equation 13. Manipulating the above expression, the first claim follows.

**Part 2.** Next, we prove the constraint derived on $h_X$. To do this, we exploit the fact that $\tilde{Y}$ is independent of $N_X$, which implies the following condition (Lin, 1997):

$$\frac{\partial^2}{\partial \tilde{y} \partial n_x} \log p(\tilde{y}, n_x) = 0, \tag{14}$$

for any $(\tilde{y}, n_x)$. According to equations equation 7, equation 8, we have that:

$$\tilde{Y} = h_Y(\tilde{X}) + N_Y,$$
$$N_X = \tilde{X} - h_X(\tilde{Y}),$$

such that we can define an invertible map $\Phi : (\tilde{y}, n_x) \mapsto (\tilde{x}, n_Y)$. It is easy to show that the Jacobian of the transformation has determinant $|J_\Phi| = 1$, such that

$$p(\tilde{y}, n_Y) = p(\tilde{x}, n_Y),$$

where $(\tilde{x}, n_Y) = \Phi^{-1}(\tilde{y}, n_X)$. Thus, being $\tilde{X}, N_Y$ independent random variables, we have that:

$$\log p(\tilde{y}, n_X) = \log p(\tilde{x}) + \log p(n_Y) = \xi(\tilde{x}) + \nu_Y(n_Y).$$

Given that $\tilde{X} = h_X(\tilde{Y}) + N_X$, we have that

$$\frac{\partial^2}{\partial \tilde{y} \partial \tilde{n}_X} \log p(\tilde{x}) = \xi'' h_X',$$

while $N_Y = \tilde{Y} - h_Y(\tilde{X})$ implies

$$\frac{\partial^2}{\partial \tilde{y} \partial \tilde{n}_X} \log p(n_Y) = -\nu_Y'' h_Y' + \nu_Y'' h_X'(h_Y')^2 - \nu_Y' h_X' h_Y'',$$

such that

$$\log p(\tilde{x}, n_Y) = \xi'' h_X' + -\nu_Y'' h_Y' + \nu_Y'' h_X'(h_Y')^2 - \nu_Y' h_X' h_Y'',$$

which must be equal to zero, being equal to the LHS of equation 14. Thus, we conclude that

$$\frac{1}{h_X'} = \frac{\xi'' + \nu_Y''(h_Y')^2 - \nu_Y' h_Y''}{\nu_Y'' h_Y'},$$

proving the claim. $\qquad\square$

### E.1 Proof of Proposition 1

*Proof.* Under the hypothesis that equations equation 5, equation 6 hold, i.e. when the data generating process satisfy both a forward and a backward model, by Theorem 2 we have that:

$$\xi'''(\tilde{x}) = \xi''(\tilde{x}) G(\tilde{x}, \tilde{y}) + H(\tilde{x}, \tilde{y}), \tag{15}$$

where

$$G(\tilde{x}, \tilde{y}) = \left( \frac{h_Y''}{h_Y'} - \frac{\nu_Y''' h_Y'}{\nu_Y''} \right),$$

$$H(\tilde{x}, \tilde{y}) = \frac{\nu_Y''' \nu_Y' h_Y'' h_Y'}{\nu_Y''} - \frac{\nu_Y'(h_Y'')^2}{h_Y'} - 2\nu_Y'' h_Y'' h_Y' + \nu_Y' h_Y'''.$$

Define $z := \xi'''$, such that the above equation can be written as $z'(\tilde{x}) = z(\tilde{x})G(\tilde{x}, \tilde{y}) + H(\tilde{x}, \tilde{y})$. given that such function $z$ exists, it is given by:

$$z(\tilde{x}) = z(\tilde{x}_0) e^{\int_{\tilde{x}_0}^{\tilde{x}} G(t,y)dt} + \int_{\tilde{x}_0}^{\tilde{x}} e^{\int_{\tilde{t}}^{\tilde{x}} G(t,y)dt} H(\hat{t}, y) d\hat{t}. \tag{16}$$

Let $\tilde{y}$ such that $\nu_Y''(\tilde{y} - h_Y(\tilde{x}))h_Y'(\tilde{x}) \neq 0$ holds for all but countable values of $\tilde{x}$. Then, $z$ is determined by $z(\tilde{x}_0)$, as we can extend equation equation 16 to all the remaining points. The set of all functions $\xi$ satisfying the differential equation equation 15 is a 3-dimensional affine space, as fixing $\xi(\tilde{x}_0), \xi''(\tilde{x}_0), \xi''(\tilde{x}_0)$ for some point $\tilde{x}_0$ completely determines the solution $\xi$. Moreover, given $\nu_Y, h_X, h_Y$ fixed, $\xi''$ is specified by equation 10 of theorem 2, which implies:

$$\xi'' = \frac{\nu_Y'' h_Y'}{h_X'} + \nu_Y' h_Y'' - \nu_Y''(h_Y')^2,$$

which confines $\xi$ solutions of equation 15 to a 2-dimensional affine space. $\qquad\square$

