# OpenReview forum: "Demystifying amortized causal discovery with transformers"
_TMLR — Accepted by TMLR_

### Review · Reviewer_gJrK · 2025-09-10

**Summary Of Contributions:**

1. The paper’s main contribution lies in connecting the emerging paradigm of transformer-based causal discovery with the foundational principles of classical identifiability theory. It helps explain why these seemingly black-box models work by arguing that the training data distribution effectively serves as an implicit set of assumptions (a prior).
2. The paper also offers a clear and actionable insight: training on a mixture of datasets generated from different, individually identifiable causal models (e.g., ANM, PNL) significantly enhances the model’s out-of-distribution generalization.
3. From my perspective, the core message is that when using amortized causal discovery methods, one should incorporate as much diversity as possible into the training datasets by drawing from more complex and identifiable data-generating processes. While this idea is, in essence, quite intuitive and straightforward, the authors provide value by formalizing it and presenting it in a clear and structured way.

**Additional Comments:**

Here are some limitations of this paper:

1.The paper’s central claim, that transformers can learn to encode identifiability assumptions from training data, is presented as an informal hypothesis that is then empirically supported, rather than formally proven. While this is acceptable given the paper’s empirical emphasis, it should be clearly stated that the contribution is not a theoretical guarantee.

2.The investigation focuses exclusively on a single architecture, CSIvA. However, the conclusions are phrased in broader terms as if they apply to amortized causal discovery with transformers in general. To avoid overgeneralization, the authors are encouraged to adjust the framing, particularly in the title and abstract, so that it better reflects the narrower scope of their empirical findings.

**Audience:**

Yes

**Audience Explanation:**

Because it addresses a timely, interesting and also important problem in the new paradigm namely amortized causal discovery. By connecting transformer-based amortized causal discovery with the classical principles of identifiability, the paper provides both conceptual clarification and practical insights that can guide future research.

**Claims And Evidence:**

Yes

**Claims Explanation:**

Yes. The claims made in the submission are supported by clear and convincing empirical evidence. The experiments are well-designed, and the results consistently align with the authors’ stated hypotheses. While the central claim is framed as an informal hypothesis rather than a formal proof, the empirical validation is sufficient and appropriate for the scope of the paper.

**Requested Changes:**

- Include the linear Gaussian model with equal variances in Section 3.1, as it is also identifiable under the ANM assumptions.

- Try to evaluate a second model architecture. To strengthen the generality of the findings, validate key results on another transformer-based model in addition to csiva. This would demonstrate that the conclusions are not limited to a single architecture.

- Discuss high-dimensional and complex settings. When extending the method, it would be valuable to include a discussion comparing your approach with methods specifically designed to handle complex mixture mechanisms, such as Causal Discovery with Mixed Linear and Nonlinear Additive Noise Models: A Scalable Approach. Such a comparison would help to better contextualize the limitations of the current work and to clarify promising directions for future research.

---

> ### Author Response · Authors · 2025-11-07
>
> We thank the reviewer for the positive feedback and constructive suggestions. Below, we address the points that were raised in the review.
>
> ---
>
> **1) On clarifying the empirical nature of the contribution.**
>
> Right after our hypothesis, we state
>
> > Notably, if this hypothesis is verified, we can analyse when CSIvA is expected or not to work well; the remainder of this work empirically studies this claim
> >
>
> To add an explicit remark about the experimental nature of our contribution, **we propose to additionally modify the first point in the bullet list of our contributions**, in the introduction section: in particular, we will write “We **empirically** show […]”.
>
> ---
>
> **2) On title rephrasing and experiments with other architectures**
>
> To justify the choice of our title, we note that CSiVA is a plain transformer (with additional attention between datapoints) that maps data to a graph. Other methods, instead, rely on different algorithmic choices (Lopez-Paz et al., Lippe et al., Li et al., Lorch et al.). In this sense, **our title is explicit about the limit of our contributions to amortized causal discovery *with transformers*,** rather than making claims for general amortized causal discovery. For this reason, we would argue that the title aligns with the scope of the paper and the experimental setup.
>
> ---
>
> **3) On abstract rephrasing.**
>
> In our abstract, we explicitly confine our contributions to CSiVA upfront, writing “In this work, **we analyze CSIvA** (Ke et al., 2023b) on bivariate causal models”. However, in line with the reviewer’s suggestion, **we agree that the last sentence of the abstract can benefit from a review that better reflects the scope of our findings**: for this, we propose the following adjustment:
>
> “Overall, we find that amortized causal discovery *with transformers* still adheres to identifiability theory […]”
>
> ---
>
> **4) Discussion about multivariate causal discovery**
>
> For a connection of our findings on bivariate graphs to high-dimensional settings, we **point the reviewer to the content of Appendix C**: we discuss results from Peters et al., showing that *identifiability* of a multivariate graph is verified if all bivariate subgraphs are identifiable. In this sense, **theory says that it is safe to extrapolate conclusions about identifiability from bivariate to multivariate settings**. Although our paper is not theoretical, if we agree that our empirical results in the main text can be explained by the theory of identifiability, the same extrapolation is well motivated.
>
> ---
>
> **5) Linear Gaussian with equal variance**
>
> **We commit to adding a discussion** on linear Gaussian with equal variance models in Section 3.1; we agree that this is pertinent and a valuable addition.
>
> ---
>
> **6)** **Comparison with the literature on mixed linear-nonlinear ANM**
>
> We take on the reviewer’s suggestion to include a discussion about existing work on *non*-amortized causal discovery with theoretical guarantees on ANMs (both linear and nonlinear). We’ll include a discussion about Liu et al. (2024) (suggested by the reviewer) and NoGAM, which also covers mixed linear and nonlinear ANMs.
>
> ---
>
> **Summary**. We will better highlight the *empirical* nature of our contributions in the introduction (point 1), on top of what is already in the paper (e.g. explicit mention that our study is *empirical* after the main motivating hypothesis). In reference to our title, we discuss why it reflects the limit of our analysis to *transformers* for amortized CD, avoiding more general claims. We commit to making this even more explicit in the abstract (points 2, 3). For a discussion on multivariate CD, we point to Appendix C (point 4). Finally, we commit to mentioning linear Gaussian models with equal variance and to discussing mixed linear-nonlinear ANM (points 5, 6).
>
> ---
>
> We renew our gratitude to the reviewer, as we believe that implementing their feedback will improve the quality of the paper.

---

> > ### Comment · Reviewer_gJrK · 2025-11-08
> > **Response to the comment**
> >
> > Thanks for the detailed response. I have no more questions.

---

> ### Comment · Action_Editor_e7eo · 2025-11-20
> **Revised manuscript uploading**
>
> Dear authors,
>
> In the response above, you have indicated some upcoming edits to the originally submitted manuscript. For example, discussion on linear Gaussian models. When are you going to upload the revised version. As far as I can see only the initial submission is currently available.
>
> Thank you!
>
> --
> AE

---

### Review · Reviewer_kUsp · 2025-09-16

**Summary Of Contributions:**

This paper offers a careful theoretical and empirical analysis of amortized causal discovery methods, with a particular focus on CSIvA. The authors’ main contributions are threefold:

1. Empirical characterization of out-of-distribution generalization limits: They show convincingly that CSIvA does not generalize to classes of SCMs not included in training, particularly when mechanisms differ.
2. Clarification of the dependence on training classes: They demonstrate that the SCM classes a model can correctly identify are precisely those it was trained on, in regimes where those families are identifiable. This ties amortized causal discovery performance to identifiability theory.
3. Theoretical justification for training on mixtures: By repurposing identifiability results for post-additive noise models, they argue that mixtures of SCM classes dramatically improve generalization, and they support this claim with strong empirical evidence.

Together these contributions provide important clarity on what amortized causal discovery can and cannot achieve, unifying and demystifying previous work in the area.

**Additional Comments:**

Nice work! Looking forward to your reply.

**Audience:**

Yes

**Audience Explanation:**

I also answered “yes” here. The findings are of clear relevance to multiple communities in the TMLR readership:
1) Causal discovery: The results clarify when amortized approaches can be expected to succeed, and when identifiability barriers persist.
2) Causal ML and causal representation learning: The analysis connects supervised learning methods like CSIvA to classical identifiability theory bridging two fields.
3) Applied researchers: Anyone considering amortized causal discovery in practice gains an immediate understanding of what kinds of training/test settings are viable.

Because amortized causal discovery has attracted growing interest in recent years, this paper’s clarifications and demystifications are timely and important.

**Claims And Evidence:**

Yes

**Claims Explanation:**

I answered “yes” here because the evidence is both accurate and convincing. The authors back each claim with clear experimental design:
1) Large synthetic datasets (15k datasets with 1.5k samples each) generated under precisely defined SCM classes.
2) Rigorous evaluation with structural Hamming distance (SHD), baselined against random guessing.
3) Replication with three random initializations and 95% confidence intervals reported.

The empirical results directly support each claim: Figures 2a/b establish the OOD generalization failure; Example 2 illustrates how identifiability drives behavior; Figures 4 and 5 clearly demonstrate the benefit of training on mixtures of SCMs. In addition, the theoretical repurposing of Hoyer et al.’s identifiability result for post-ANMs provides a sound explanation for why the empirical findings make sense. Overall, the claims are neither overstated nor speculative; they are carefully aligned with the provided evidence.

**Requested Changes:**

I have a few suggestions for clarity and polish:

Section 3.2 wording: Replace “Our arguments of Example 1” with “The claims made regarding Example 1.” Clarify the final sentence (“Additionally, it…”): specify that CSIvA can be explicitly trained to recognize assumptions, and give a concrete example of what assumptions/contexts are meant.

Section 4.1 wording: Instead of “Each dataset is generated according to a single class of SCMs,” write: “When studying a given SCM class (e.g. nonlinear-Gaussian), we generate each dataset from a single SCM instance sampled from that class.”

Section 4.2 wording: Remove “in the first place” in “can not be found in Ke et al. (2023b) in the first place.”
Correct “that those” → “than those” in “different mechanism types and noise distributions than those it was trained on.”

Section 4.3 wording: In Example 2, define “forward” and “backward” models before use. For example: “a backward linear model (i.e., Y → X with linear f()).” Also consider changing “encompass” in the posed question to “correctly classify.”

General: In the hypothesis box, consider a more precise phrasing, e.g.: “CSIvA generalizes to the SCM families it was trained on, in the regimes where those families are identifiable.” This would sharpen the main insight while remaining informal.

---

> ### Author Response · Authors · 2025-11-05
>
> **We thank the reviewer** for their positive feedback, as well as the constructive suggestions. Regarding the requested changes in terms of wordings, we commit to implement them: we agree that they would improve the clarity (and sometimes, correctness) of the writing.
>
> **Regarding the phrasing of our hypothesis**, we take on the reviewer’s suggestion and propose the following adjustment: “At test time, CSIvA generalizes to datasets drawn from families of SCMs that are represented in the training distribution, if those families are identifiable from observational data.” Looking forward to your opinion!

---

> > ### Comment · Reviewer_kUsp · 2025-11-13
> >
> > That phrasing of the hypothesis sounds great!

---

### Review · Reviewer_kcRi · 2025-10-14

**Summary Of Contributions:**

The authors highlight a tension between the limitations of identification of graphs from observed data (i.e. that bivariate graphs are not a priori identifiable from observed data alone)  and the empirical evidence, which highlights that in practice many causal discovery methods perform well on synthetic benchmarks.  The authors investigate this phenomenon to understand why and under what conditions it is possible to identify a graph in spite of the lack of identification guarantees associated with the class of algorithms they consider, in particular the CSIvA architecture.

Perhaps unsurprisingly, they show that suitable assumptions much be encoded in the training distribution for the successful identification of graphs in the test data. They stick to bivariate graphs but lean on the intuition that if one can identify a bivariate graph, one can identify a multivariate graph by the same logic (by extending the guarantees to all bivariate subgraphs).

**Audience:**

Yes

**Audience Explanation:**

Overall the paper concerns an important topic which is the limits of these amortized approaches to causal discovery.

**Claims And Evidence:**

Yes

**Claims Explanation:**

For evaluation the authors use a sample of 15k synthetic datasets each consisting 1500 observations. They standardise variance of the two variables (a detail which is often overlooked) to avoid any possible shortcuts for inference.

Also as expected, they find that CSIvA can generalise from train to test within distribution, but not out of distribution. Interesting this failure to generalise applies to both the mechanism mapping X to Y (or vice versa) as well as the choice of noise distribution.

Motivated by the confirmation that CSIvA follows what one might expect from identifiability theory, they explore whether a model could be training on multiple (or in principle all) identifiable causal structures. One interesting example which suggests that this might be difficult relates to particular combinations of examples which result in a failure to identify the correct causal graph.

**Requested Changes:**

Major:

It’s good to have this kind of empirical validation in ‘one place’ but I found  (a) the exploration rather simple and (b) the results rather obvious. Neither of these are reasons for rejecting a paper per se, but nonetheless, as I was reading the paper I kept waiting for the ‘Aha!’ moment, which never really arrived. If I had to paraphrase the conclusion, its that these amortised approaches to causal discovery provide no guarantees for identification (which we already knew), but empirically they show some reasonable performance if the examples they are trained on look like the examples they are tested on in terms of mechanism and distribution (which we also kinda knew). If the authors would argue that we really didn’t understand this already then maybe they can make a strong case by ‘identifying’ (excuse the pun) the failure of prior literature. This might make the work more compelling.

I don’t want to be unfair to the authors, because the paper was well written, well structured, and easy to follow. I’d be interested to hear what the other reviewers think in terms of possible suggestions for making the contribution a bit more substantial.

Minor:

Figure 1 caption says that CSIvA is ‘generalising well’, suggest this is just clarified in the text to be in-distirbution generalisation so that its super clear when scanning through (I had to double check this was within-distribution generalisation and not OOD). Figure 2’s caption is clearer, for example.

---

> ### Author Response · Authors · 2025-11-05
>
> We thank the reviewer for the time dedicated to our work and for their encouraging comments. Below, we provide a structured response to the reviewer’s concerns.
>
> ---
>
> **1) On “unsurprising” results and why they matter**
>
> One of the premises of amortized causal discovery comes from the influential hypothesis of Lopez Paz et al. (2015):
>
> > “…learning-based approaches to causal discovery would […] greatly reduce the need for explicitly crafting identifiability conditions a-priori.”
> >
>
> Our work ”fact-checks” this claim, and finds evidence that **amortization does not bypass identifiability**—it **moves assumptions into the training prior**. Despite lacking the excitement of an “Aha!” moment, this fact is overlooked in amortized causal discovery research. This is suggestive that our findings are not obvious for at least part of the causality community.
>
> ---
>
> **2) Our results are relevant for amortized causal effect estimation**
>
> A parallel and recent trend proposes **amortized causal** **effect estimation**: the idea is training on synthetic SCMs and aiming for zero-shot deployment on test data. Recent work from leading labs (Robertson et al., NeurIPS 2025) writes:
>
> > “We pre-train PFNs on synthetic data drawn from a wide variety of causal structures, including interventions, to predict interventional outcomes given observational data… [this] allows for the accurate estimation of causal effects without knowledge of the underlying causal graph.”
> >
>
> Similar ideas propel other works (e.g. Balazadeh et al., in Neurips 2025, and Bynum et al., 2025). Causal effect estimation is a downstream task of causal discovery (which is needed for valid adjustments): the caution that we find to be necessary in amortized causal discovery is thus necessary for causal effect estimation. In contrast, Robertson et al. suggest that identifiability constraints can be bypassed by amortized inference (akin to Lopez-Paz et al.'s claim for causal discovery). In light of this work, we believe that the **limitations imposed by classical causality theory on amortized causal inference that we find are not obvious to a part of the causality community**.
>
> ---
>
> **3) The message of the paper is not that amortized CD lacks guarantees**
>
> The reviewer summarises some our findings as follows: “If I had to paraphrase the conclusion, its that these amortised approaches to causal discovery provide no guarantees for identification” There might be a misunderstanding in the message of the paper: our work provides empirical evidence that the **identifiability guarantees of amortized CD can be predicted** pairing the classical theory of causal discovery with knowledge about the class of SCMs used during training, which acts as a prior on the test data.
>
> ---
>
> **4) Tradeoffs between amortized and classical CD.**
>
> The other key contribution of our work is that it is the first to find explicit tradeoffs between amortized and classical causal discovery:
>
> - **Amortized CD (e.g., CSIvA)** is sensitive to shifts in the mechanisms class and the noise distributions. While training on different mechanisms classes (linear, ANM, PNL) is relatively easy, it’s infeasible to train a method over SCMs covering all possible noise distributions. In this sense, amortized CD is likely always making inference OOD, when it comes to noise.
> - **Classical CD (e.g., RESIT)** needs **structural assumptions** (ANM) yet is comparatively **less sensitive to noise-family mismatch**.
>
> This trade-off is unknown in previous research, being our work the first systematic OOD generalization study for amortized CD.
>
> ---
>
> **Summary.** We argue that research on amortized causal discovery and causal effect estimation is unaware of the limitations from the classical theory of causal discovery: this is evidence that our findings are not obvious to some of the causal discovery community (points 1 and 2). Our main message is that amortized causal discovery still requires prior on the test data in order to be reliable (point 3). Moreover, our work highlights trade-offs between classical and amortized CD (point 4).
>
> **References.**
>
> Do-PFN: In-Context Learning for Causal Effect Estimation, NeurIPS 2025. Robertson et al.
>
> CausalPFN: Amortized Causal Effect Estimation via In-Context Learning, NeurIPS 2025, Balazadeh et al.
>
> Black Box Causal Inference: Effect Estimation via Meta Prediction, Bynum et al. (2025), preprint

---

> > ### Comment · Reviewer_kcRi · 2025-11-06
> > **Response to Comment**
> >
> > Thanks a lot to the authors for taking the criticisms in the spirit in which they were intended (i.e. hopefully constructively), and for clarifying the key contributions of the paper. I have to say I'm not sure this changes a huge deal for me ("amortization does not bypass identifiability—it moves assumptions into the training prior" still seems like a given, and very similar ideas have been also discussed/accepted in parallel terms in latent variable modelling for at least 6 years - thinking of Dai and Wipf 2019 or Locatello et al. 2019 in the context of inferring causal structure in generative models), but I appreciate the authors highlighting the potential lack of awareness in this specific literature/community. Perhaps I didn't appreciate this when reading it the first time. If the authors feel there is anything they could emphasize to this effect then I imagine it might be worthwhile for other readers (at least ones like myself, to the extent that they do/don't exist...).
> >
> > In any case I am leaning firmly positively for the paper.

---

### Author Response · Authors · 2025-10-22
**Extension of the discussion period**

Dear reviewers,

I am writing to you on behalf of the authors of the paper. Thank you for the time dedicated to our work and the constructive reviews. We sincerely appreciate that.

We would like to notify you that the action editor has agreed, upon our request, to extend the discussion period until November 10th. This is because right now we (the authors) can not dedicate the time needed to properly address the reviews, due to the overlap with other work duties. We thank you all for your understanding, and apologise for any inconvenience.

---

### Author Response · Authors · 2025-11-05
**General comment**

We thank all reviewers for their time, constructive feedback, and encouraging comments. There is unanimous agreement that our work provides **accurate, convincing, and clear evidence** for the claims, and its content is **interesting for at least some of the TMLR audience**. Moreover, there is agreement that:

1. Our work treats an **interesting topic** (all reviewers)
2. Content of our work is **timely and important** (kUsp, gJrK)
3. The paper is **well written and structured** (kcRi, kUsp)

Potential criticisms from each reviewer are addressed in the individual responses. However, we use this space for one remark of general interest: the main concern about our work comes from Reviewer kcRi, arguing that our findings are unsurprising in the light of the classical theory of causal discovery; contextually, Reviewer gJrK highlights that the evidence in our paper is empirical, requiring care in extrapolating theoretical conclusions from it. In this sense, one reviewer argues that the theory underlying the experimental findings is self evident, the other that interpretation of the theory underlying the experiments should be done with care: this **“tension” between views is paradigmatic of the fact that the theoretical implications of amortized causal discovery are not obvious**, and require research—our work is the first to reduce this gap, as agreed by reviewers kUsp and gJrK.

---

### Decision · Action_Editor_e7eo · 2025-11-20

**Recommendation:** Accept with minor revision

**Additional Comments:**

The paper does not fall in any of the 3 categories going by how they are defined here https://jmlr.org/tmlr/editorial-policies.html#certifications.

**Audience:**

Yes

**Audience Explanation:**

The paper puts together a solid evidence that amortized causal discovery works within the training distribution and as such would be useful for the parties arguing for either way.

**Claims And Evidence:**

Yes

**Claims Explanation:**

All three reviewers agreed with that.